# Rethinking Approximate Gaussian Inference in Classification

**Bálint Mucsányi**[*] **Nathaël Da Costa**[*] **Philipp Hennig**

Tübingen AI Center
University of Tübingen

## Abstract

In classification tasks, softmax functions are ubiquitously used as output activations to produce predictive probabilities. Such outputs only capture aleatoric uncertainty. To capture epistemic uncertainty, approximate Gaussian inference methods have been proposed. We develop a common formalism to describe such methods, which we view as outputting Gaussian distributions over the logit space. Predictives are then obtained as the expectations of the Gaussian distributions pushed forward through the softmax. However, such softmax Gaussian integrals cannot be solved analytically, and Monte Carlo (MC) approximations can be costly and noisy. We propose to replace the softmax activation by element-wise normCDF or sigmoid, which allows for the accurate sampling-free approximation of predictives. This also enables the approximation of the Gaussian pushforwards by Dirichlet distributions with moment matching. This approach entirely eliminates the runtime and memory overhead associated with MC sampling. We evaluate it combined with several approximate Gaussian inference methods (Laplace, HET, SNGP) on large- and small-scale datasets (ImageNet, CIFAR-100, CIFAR-10), demonstrating improved uncertainty quantification capabilities compared to softmax MC sampling. Our code is available at github.com/bmucsanyi/probit.

## 1 Introduction

Given an input space $\mathcal{X}$, $C$ classes and training data $(x_n, c_n)_{n=1}^N \subset \mathcal{X} \times \{1, \ldots, C\}$, the goal of probabilistic classification is to learn a function $\boldsymbol{h} \colon \mathcal{X} \to \Delta^{C-1}$, where

$$\Delta^{C-1} = \{(p_1, \ldots, p_C) \in [0,1]^C : p_1 + \cdots + p_C = 1\} \tag{1}$$

is the $(C-1)$-dimensional probability simplex. The model $\boldsymbol{h}$ outputs the probability of inputs $x \in \mathcal{X}$ belonging to each class, $h_c(x) = p(c \mid x)$. To learn such a map to the simplex, $\boldsymbol{h}$ is typically defined as a composition

$$\boldsymbol{h} \colon \mathcal{X} \xrightarrow{\ \boldsymbol{f}\ } \mathbb{R}^C \xrightarrow{\ \textbf{softmax}\ } \Delta^{C-1} \tag{2}$$

where a function $\boldsymbol{f}$, learned from the training data, is mapped through the softmax activation function. In this case, $\mathbb{R}^C$ is called the *logit space*, and, for $x \in \mathcal{X}$, $\boldsymbol{f}(x)$ is a *logit*.

With the usual assumption of i.i.d. data, the likelihood under the model $\boldsymbol{h}$ is given by

$$p((c_n)_{n=1}^N \mid (x_n)_{n=1}^N) = \prod_{n=1}^N p(c_n \mid x_n) = \prod_{n=1}^N \prod_{c=1}^C h_c(x_n)^{\delta_{c_n, c}}, \tag{3}$$

---

[*]Equal contribution.

39th Conference on Neural Information Processing Systems (NeurIPS 2025).

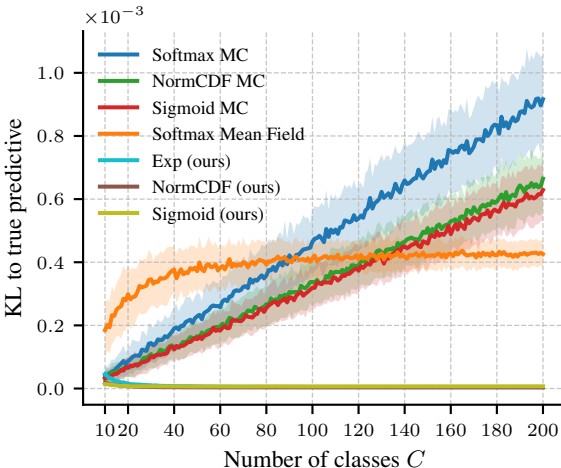

Figure 1: Mean KL divergence between the 'true' predictive (approximated with a 10000 sample MC approximation) and different predictive approximations on a synthetic data set. This synthetic dataset consists of 100 logit means $\boldsymbol{\mu}$ and standard deviations $\boldsymbol{\sigma}$, whose components are i.i.d. uniformly distributed. For a fair comparison we scale the logit dataset for the different activations to match the activations to first order, so that $\mu_c \in [-1, 1]$, $\sigma_c \in [0, 1]$ for sigmoid, $\mu_c \in [-1/2 - \log 2, 1/2 - \log 2]$, $\sigma_c \in [0, 1/2]$ for softmax, $\mu_c \in [-\sqrt{\pi/8}, \sqrt{\pi/8}]$, $\sigma_c \in [0, \pi/8]$ for normCDF. The MC approximations are capped at a fixed computational budget of 10000 class samples, i.e. $\lceil 10000/C \rceil$ samples (see also Appendix C.1).

where $\delta$ is the Kronecker delta. This yields a natural loss function for classification problems, the cross-entropy (CE) loss

$$\mathcal{L}((x_n, c_n)_{n=1}^N) = -\log(p((c_n)_{n=1}^N \mid (x_n)_{n=1}^N)) = -\sum_{n=1}^N \sum_{c=1}^C \delta_{c_n, c} \log(h_c(x_n)). \qquad (4)$$

For typical models $\boldsymbol{h}$ trained through such approximate maximum likelihood estimation, an output probability vector $\boldsymbol{h}(x) \in \Delta^{C-1}$ should be interpreted as an estimate of $p_{\text{gen}}(\boldsymbol{y} \mid x)$, the probability under the generative model. Notably, the output probability does not take into account the uncertainty of the model due to the finite nature of the data. In the uncertainty quantification formalism, such models can only estimate *aleatoric uncertainty* and disregard *epistemic uncertainty* [21].

The probabilistic way to capture epistemic uncertainty is to require the model to output a *second-order distribution* (that is 'a distribution over probability distributions'). Namely, for each input $x \in \mathcal{X}$, the model should output a probability measure over the simplex $\text{h}(x) \in \mathcal{P}(\Delta^{C-1})$, as opposed to a point estimate in the simplex $\boldsymbol{h}(x) \in \Delta^{C-1}$ [46, 6].

A number of methods for such distributional uncertainty quantification in classification rely on *approximate Gaussian inference* in logit space. That is, h is written as a composition

$$\text{h} \colon \mathcal{X} \xrightarrow{\ \text{f}\ } \mathcal{G}(\mathbb{R}^C) \xrightarrow{\ \textbf{softmax}_* \ } \mathcal{P}(\Delta^{C-1}) \qquad (5)$$

where f is learned from the training data, $\mathcal{G}(\mathbb{R}^C)$ is the set of Gaussian measures on the logit space $\mathbb{R}^C$, and $\textbf{softmax}_*$ pushes forward the Gaussian probability measures through the softmax. For example, Heteroscedastic Classifiers (HET) [7] learn the output means and covariances of f explicitly with a neural network. In linearised Laplace approximations [11], only the means are learned explicitly with a neural network, while the covariances are obtained post-hoc from that neural network by approximating its parameter posterior distribution with a Gaussian, and then pushing it forward to logit space by linearising the network in the parameters. Last-layer Laplace approximations [25] are a variant of linearised Laplace approximations where only the posterior over the last-layer parameters is Gaussian-approximated, requiring no linearisation of the network. Spectral-Normalised Gaussian processes (SNGP) [31] are neural networks with last layers that are approximately Gaussian processes, through random feature expansions and Laplace approximations.

To obtain predictive probabilities from a distributional classifier as in Eq. (5), we marginalise it out by the measure-theoretic change of variables, using $\text{h}(x) = \textbf{softmax}_* \text{f}(x)$:

$$\mathbb{E}_{\boldsymbol{P} \sim \text{h}(x)}[\boldsymbol{P}] = \int_{\Delta^{C-1}} p \, \text{h}(x)(dp) = \int_{\mathbb{R}^C} \textbf{softmax}(y) \, \text{f}(x)(dy). \qquad (6)$$

In addition to predictive probabilities, the distributional framework allows for the acquisition of other quantities of interest, such as variances, entropy, and information-theoretic decompositions of the

predictive uncertainty into aleatoric and epistemic parts [13, 41, 55]. These are usually obtained by Monte Carlo (MC) sampling from the Gaussian measure f($x$). Indeed, while the probability density of h($x$) = **softmax**$_*$ f($x$) can be obtained in closed form using a change of variables formula [2], this does not seem of practical use—even predictives cannot be computed analytically. On the other hand, given a variance, the runtime and memory cost required for an MC approximation to achieve an estimator of said variance grows linearly with the number of classes. Crucially, this computational cost is not limited to *training time*, but also to *inference time*, when the computational budget may be much more limited. This limits the application of such Gaussian inference methods in classification with a large number of classes, such as those common in computer vision and language modelling.

**Summary of Contributions:** Our work extends the classical probit approximation for the Gaussian sigmoid integral in binary classification [49, 34] in the following manner:

(i) Generalising it to the multi-class setting (Sections 3.1 and 3.2).

(ii) Analysing it theoretically (Section 3.3).

(iii) Analysing it empirically, and how it outperforms previous approximations (Fig. 1 and Section 7).

(iv) Generalising it to second-order distributional Dirichlet approximations (Section 4).

(v) Combining it with advantageous training (Sections 5 and 7).

Our proposed framework can be summarised as a 3-step recipe:

> (i) Choose an approximate Gaussian inference method that outputs logit means and covariances.
>
> (ii) Choose an output activation that yields a closed-form Gaussian integral (Section 3).
>
> (iii) At inference time, use our closed-form formulas (Section 3 for the predictives, Section 4 for other quantities of interest).

## 2   Related Work

In uncertainty quantification, methods that output logit Gaussians, which we refer to as *approximate Gaussian inference* methods, have become one of the dominant paradigms in uncertainty quantification [7, 11, 25, 31]. In classification tasks, such methods have the downside of requiring sampling at inference to transform Gaussian distributions into predictive probability vectors.

In parallel, prior work has developed ways to propagate probabilistic uncertainty through neural networks [16]. Several works develop analytic propagation of covariances up to the logits [56, 29]; our method complements these by addressing the propagation of the logit-space Gaussian distributions onto the probability simplex. Few works address this last step. The Laplace bridge [19] proposes a map from Gaussians to Dirichlets on the probability simplex. However, this approach has no guarantees on the quality of its predictives, and it severely underperforms Monte Carlo sampling. Other approaches include the mean-field approximation to the Gaussian softmax integral [32] and the approximations of Shekhovtsov and Flach [48]. In this work, we propose approximate predictives that outperform existing closed-form predictive techniques and provide second-order Dirichlet distributions for other closed-form quantities of interest. Furthermore, this is the first work to provide a theoretical analysis of such an approximation.

Other works have also proposed various activations for classification to replace the softmax. Particularly relevant is [15], which composes element-wise sigmoids with normalisation and utilises auxiliary-variable augmentation to obtain closed-form ELBO terms during training; however, its predictives require Monte Carlo estimation. Related conjugate/augmentation-based objectives exist for stick-breaking and tree-structured multi-class GPs [30, 1], and for BNNs, a tempered-likelihood/Dirichlet perspective with a factorised-Gaussian surrogate has been explored [24]. Yet, all these approaches still require sampling or numerical quadrature at inference time. [49, 34, 25] use the probit approximation in the binary case, which we generalise to the multi-class setting. [51] approximates the softmax by providing lower bounds for point estimate training, but replacing the softmax with these bounds during training does not yield a normalised multi-class likelihood. The binary cross-entropy loss with the element-wise sigmoid activation has been reported to outperform cross-entropy in large-scale settings [54, 28]. Our work is the first to use this advantageous loss while normalising the output to obtain proper multi-class probabilistic models with closed-form predictives from logit Gaussians.

Evidential deep learning (EDL) approaches [46] offer sample-free outputs through Dirichlet distributions but have been shown to be unreliable for uncertainty estimation and disentanglement [4, 23, 44, 40]. We focus on making approximate Gaussian inference methods sample-free in a principled and performant manner; however, for completeness, we compare our approach with EDL methods in Appendix K.8.

Finally, multiple authors have explored continuous approximations to categorical distributions, which enable gradient-based optimization for discrete stochastic models through sampling, such as the Gumbel-max trick [35, 22, 36, 20]. These methods principally optimise a single categorical distribution, while the present work is concerned with constructing probability densities over the (continuous) simplex of such categorical distributions.

## 3 Gaussian Inference with Closed-Form Predictives

### 3.1 Motivation from Softmax Models

A key difficulty in obtaining exact or closed-form approximate predictive probabilities of $h(x) = \mathbf{softmax}_* f(x)$ is that pushforwards of Gaussian distributions through the softmax are intractable. The softmax function is the composition $\mathbf{softmax} = n \circ \exp$, where $n$ is the normalisation function given by $n(q) := q/(q_1 + \cdots + q_C)$ and $\exp$ is applied element-wise. Thus, $h$ can be written as the composition (see Eq. (5))

$$h \colon \mathcal{X} \xrightarrow{\ f\ } \mathcal{G}(\mathbb{R}^C) \xrightarrow{\ \mathbf{exp}_*\ } \mathcal{P}((0,\infty)^C) \xrightarrow{\ n_*\ } \mathcal{P}(\Delta^{C-1}). \tag{7}$$

Now it is straightforward to see that (Appendix A.1)

$$\int_{\mathbb{R}} \exp(y)\, \mathcal{N}(\mu, \sigma^2)(dy) = \exp\left(\mu + \frac{\sigma^2}{2}\right) \tag{8}$$

and thus

$$\mathbb{E}_{Q \sim \mathbf{exp}_* f(x)}[Q] = \int_{\mathbb{R}^C} \mathbf{exp}(y)\, f(x)(dy) = \mathbf{exp}\left(\mu(x) + \frac{\sigma^2(x)}{2}\right) \tag{9}$$

where $\mu(x)$ and $\sigma^2(x)$ are, respectively, the mean and the diagonal of the covariance of $f(x)$, and the vector division is element-wise. Note that no approximations have been made so far.

One can use this to approximate the model's predictive in closed form:

$$\mathbb{E}_{P \sim h(x)}[P] = \mathbb{E}_{Q \sim \mathbf{exp}_* f(x)}\left[\frac{Q}{\sum_{c=1}^C Q_c}\right] \approx \frac{\mathbb{E}_{Q \sim \mathbf{exp}_* f(x)}[Q]}{\mathbb{E}_{Q \sim \mathbf{exp}_* f(x)}\left[\sum_{c=1}^C Q_c\right]}$$
$$= \frac{\mathbf{exp}\left(\mu(x) + \frac{\sigma^2(x)}{2}\right)}{\sum_{c=1}^C \exp\left(\mu_c(x) + \frac{\sigma_c^2(x)}{2}\right)}. \tag{10}$$

### 3.2 The General Framework

The previous argumentation can be applied to a general family of output activation functions: suppose the output activation is of the form $n \circ \varphi$, where $\varphi$ is the element-wise application of some activation function $\varphi \colon \mathbb{R} \to (0,\infty)$. Then we have (see Eq. (7))

$$h \colon \mathcal{X} \xrightarrow{\ f\ } \mathcal{G}(\mathbb{R}^C) \xrightarrow{\ \varphi_*\ } \mathcal{P}((0,\infty)^C) \xrightarrow{\ n_*\ } \mathcal{P}(\Delta^{C-1}). \tag{11}$$

Solving the one-dimensional integrals $\int_{\mathbb{R}} \varphi(y)\, \mathcal{N}(\mu, \sigma^2)(dy)$ (see Eq. (8)) allows us to obtain the approximate predictive (see Eq. (10))

$$\mathbb{E}_{P \sim h(x)}[P] \approx \frac{\mathbb{E}_{Q \sim \varphi_* f(x)}[Q]}{\sum_{c=1}^C \mathbb{E}_{Q_c \sim \varphi_* f_c(x)}[Q_c]}. \tag{12}$$

For instance, taking $\varphi = \Phi$, the standard Normal cumulative distribution function (normCDF)[2], we have by a classical result (Appendix A.2)

$$\int_{\mathbb{R}} \Phi(y)\, \mathcal{N}(\mu, \sigma^2)(dy) = \Phi\left(\frac{\mu}{\sqrt{1+\sigma^2}}\right). \tag{13}$$

---

[2]The resulting model is distinct from a multinomial probit model, see Appendix B.

and thus

$$\mathbb{E}_{\boldsymbol{P}\sim\mathrm{h}(x)}[\boldsymbol{P}] \approx \frac{\boldsymbol{\Phi}\left(\frac{\boldsymbol{\mu}(x)}{\sqrt{1+\boldsymbol{\sigma}^2(x)}}\right)}{\sum_{c=1}^{C}\Phi\left(\frac{\mu_c(x)}{\sqrt{1+\sigma_c^2(x)}}\right)}. \tag{14}$$

Taking instead $\varphi = \rho$, the logistic sigmoid, we approximate Eq. (13) using the probit approximation (Appendix A.3),

$$\int_{\mathbb{R}} \rho(y)\,\mathcal{N}(\mu,\sigma^2)(dy) \approx \rho\left(\frac{\mu}{\sqrt{1+\frac{\pi}{8}\sigma^2}}\right) \tag{15}$$

yielding in this case

$$\mathbb{E}_{\boldsymbol{P}\sim\mathrm{h}(x)}[\boldsymbol{P}] \approx \frac{\boldsymbol{\rho}\left(\frac{\boldsymbol{\mu}(x)}{\sqrt{1+\frac{\pi}{8}\boldsymbol{\sigma}^2(x)}}\right)}{\sum_{c=1}^{C}\rho\left(\frac{\mu_c(x)}{\sqrt{1+\frac{\pi}{8}\sigma_c^2(x)}}\right)}. \tag{16}$$

### 3.3 Theoretical Analysis of the Approximation's Quality

In Fig. 1, we see empirically that the quality of our predictive approximations outperforms existing predictive approximations on synthetic data. We now explore the quality of our predictive approximations theoretically. We conduct the analysis for the case $\varphi = \rho$. Let

$$q = q(\mu,\sigma^2) := \mathbb{E}_{Q\sim\rho_*\mathcal{N}(\mu,\sigma^2)}[Q] \approx \rho\left(\frac{\mu}{\sqrt{1+\frac{\pi}{8}\sigma^2}}\right) =: \hat{q}(\mu,\sigma^2) = \hat{q}, \tag{17}$$

where $\hat{q}$ is the probit approximation (Appendix A.3). Moreover we write $q_c := q(\mu_c,\sigma_c^2)$, $\hat{q}_c := \hat{q}(\mu_c,\sigma_c^2)$ where $(\mu_c,\sigma_c^2)$ is the logit mean-variance pair for class $c$, $\boldsymbol{p}$ for the true predictive and $\hat{\boldsymbol{p}}$ for our approximate predictive, i.e.

$$\boldsymbol{p} := \mathbb{E}_{\boldsymbol{P}\sim(\boldsymbol{n}\circ\boldsymbol{\rho})_*\mathcal{N}(\boldsymbol{\mu},\boldsymbol{\Sigma})}[\boldsymbol{P}], \quad \hat{\boldsymbol{p}} := \frac{\hat{q}_c}{\sum_{c'=1}^{C}\hat{q}_{c'}}, \tag{18}$$

where $\boldsymbol{\Sigma}$ has $\boldsymbol{\sigma}$ as diagonal. To analyse the quality of the approximation, we thus want to bound $\mathrm{D}_{\mathrm{KL}}(\boldsymbol{p},\hat{\boldsymbol{p}})$. Clearly, when all the $Q_c$ are deterministic (i.e. constant random variables), where $\boldsymbol{Q} \sim \boldsymbol{\varphi}_*\mathrm{f}(x)$, we have $\boldsymbol{p} = \hat{\boldsymbol{p}}$. Alternatively, from the derivation Eq. (10), we see that in the limiting case Eq. (12) is exact when $\sum_{c=1}^{C}Q_c$ is deterministic. This is a weaker assumption than requiring all $Q_c$ to be deterministic individually. In fact, we can derive a rate of decay of $\mathrm{D}_{\mathrm{KL}}(\boldsymbol{p},\hat{\boldsymbol{p}})$ in terms of $\mathrm{Var}\left(\sum_{c=1}^{C}Q_c\right)$:

**Theorem 3.1.** *Suppose the means and variances $(\mu_c,\sigma_c^2)$ lie in some compact set $\mathcal{K} \subset \mathbb{R} \times [0,\infty)$ for each class $c$. Then*

$$\mathrm{D}_{\mathrm{KL}}(\boldsymbol{p},\hat{\boldsymbol{p}}) \leq M(\mathcal{K}) + O\left(\mathrm{Var}\left(\sum_{c=1}^{C}Q_c\right)\right) \tag{19}$$

*as $\mathrm{Var}\left(\sum_{c=1}^{C}Q_c\right) \to 0$ and $M(\mathcal{K}) \in [0,\infty]$ some constant which only depends on $\mathcal{K}$, not on $C$, such that $M(\mathcal{K}) < \infty$ when $\sup_{(\mu,\sigma^2)\in\mathcal{K}}\frac{q-\hat{q}}{q} < 1$, and $M(\mathcal{K}) = 0$ when $q = \hat{q}$ for all $(\mu,\sigma^2) \in \mathcal{K}$.*

Appendix C.2 and Theorem C.2 provide a formula for $M(\mathcal{K})$ and a proof. We also conduct an empirical investigation of whether the assumptions made in the theorem hold in practice, namely whether the logit means and variances lie in a compact set, and whether $M(\mathcal{K}) < \infty$.

The term $M(\mathcal{K})$ in Eq. (19) results from the error due to the probit approximation, while the $O$ term is due to the splitting of the expectation into the numerator and denominator in the derivation Eq. (10). Importantly, Theorem 3.1 states that the former does not increase with the number of classes. For the latter, requiring $\mathrm{Var}\left(\sum_{c=1}^{C}Q_c\right) \to 0$ is weaker than requiring $\mathrm{Var}(Q_c) \to 0$ for all $c$ (see Eq. (63) in Appendix C.2); but we show in Appendix C.2 that assuming $\mathrm{Var}(Q_c) \to 0$ for all $c$ gives almost twice as strong a decay rate in $\mathrm{D}_{\mathrm{KL}}$. We include this additional result in Theorem C.2.

# 4 Distributional Approximations of the Gaussian Pushforwards

## 4.1 Dirichlet Matching

While in many applications, predictive probabilities are all that is needed from a model, we set out to build a model that outputs a tractable probability distribution over the simplex $h(x) \in \mathcal{P}(\Delta^{C-1})$. As previously discussed, such a distribution allows, for instance, closed-form decompositions of aleatoric and epistemic uncertainties.

The formulation of Eq. (11) does not yet allow this. However, we will now see that one can obtain good Dirichlet approximations by constructing a tractable approximation of Eq. (11) of the form

$$h \colon \mathcal{X} \xrightarrow{\quad f \quad} \mathcal{G}(\mathbb{R}^C) \xrightarrow{\quad a \quad} \mathcal{D}(\Delta^{C-1}), \tag{20}$$

where $\mathcal{D}(\Delta^{C-1})$ is the space of Dirichlet distributions on $\Delta^{C-1}$. The family of Dirichlets presents an ideal choice of distributions over the simplex as, for such distributions, all quantities of interest are closed-form (see Appendix F). The map a is constructed by moment matching: we match the first two moments of the Dirichlet distribution to the moments of the Gaussian pushforward $\varphi_* f(x)$.

Conveniently, like the first moments described in Section 3, we can obtain second moments for $\exp$

$$\int_{\mathbb{R}} \exp(y)^2 \, \mathcal{N}(\mu, \sigma^2)(dy) = \exp(2\mu + 2\sigma^2) \tag{21}$$

and for $\Phi$ (Owen [43, Eq. 20,010.4])

$$\int_{\mathbb{R}} \Phi(y)^2 \, \mathcal{N}(\mu, \sigma^2)(dy) = \Phi\left(\frac{\mu}{\sqrt{1+\sigma^2}}\right) - 2T\left(\frac{\mu}{\sqrt{1+\sigma^2}}, \frac{1}{\sqrt{1+2\sigma^2}}\right), \tag{22}$$

where $T$ is Owen's T function. For $\rho$, we have the following approximation (Daunizeau [10, Equation 23])

$$\int_{\mathbb{R}} \rho(y)^2 \, \mathcal{N}(\mu, \sigma^2)(dy) \approx \rho\left(\frac{\mu}{\sqrt{1+\frac{\pi}{8}\sigma^2}}\right) - \frac{\rho\left(\frac{\mu}{\sqrt{1+\frac{\pi}{8}\sigma^2}}\right)\left(1 - \rho\left(\frac{\mu}{\sqrt{1+\frac{\pi}{8}\sigma^2}}\right)\right)}{\sqrt{1+\frac{\pi}{8}\sigma^2}}. \tag{23}$$

This enables us to compute the second moment of the pushforward distribution via

$$\mathbb{E}_{\boldsymbol{P} \sim \boldsymbol{n}_* \varphi_* f(x)}[\boldsymbol{P}^2] = \mathbb{E}_{\boldsymbol{Q} \sim \varphi_* f(x)}\left[\frac{\boldsymbol{Q}^2}{\left(\sum_{c=1}^C Q_c\right)^2}\right] \approx \frac{\mathbb{E}_{\boldsymbol{Q} \sim \varphi_* f(x)}[\boldsymbol{Q}^2]}{\mathbb{E}_{Q_c \sim \varphi_* f_c(x)}\left[\sum_{c=1}^C Q_c\right]^2}. \tag{24}$$

As in Eq. (12), the approximation in Eq. (24) is exact when $\sum_{c=1}^C Q_c$ is a constant random variable.

If we were to directly attempt to infer the parameters of the Dirichlet $h(x)$ from Eqs. (12) and (24), we would obtain an overparametrised system of equations with $2C$ equations and $C$ unknowns. Thus, we instead use a classical Dirichlet method of moments [38, Eqs. 19 & 23], which reduces these equations, as we will now describe.

The sum of the Dirichlet parameters $\sum_{c=1}^C \gamma_c$ may be estimated by

$$\frac{\mathbb{E}[P_c] - \mathbb{E}[P_c^2]}{\mathbb{E}[P_c^2] - \mathbb{E}[P_c]^2} \tag{25}$$

for any $1 \leq c \leq C$. So, a natural estimate of $\sum_{c=1}^C \gamma_c$ that uses all $P_c$ is the geometric mean of Eq. (25). Since the mean of the Dirichlet is $\boldsymbol{\gamma}/\sum_{c=1}^C \gamma_c$, we obtain using Eq. (25) an expression for the Dirichlet parameters

$$\left(\prod_{c=1}^C \frac{\mathbb{E}[P_c] - \mathbb{E}[P_c^2]}{\mathbb{E}[P_c^2] - \mathbb{E}[P_c]^2}\right)^{1/C} \mathbb{E}[\boldsymbol{P}]. \tag{26}$$

Hence, using Eqs. (12) and (24), we obtain the computationally feasible Dirichlet parameters

$$\boldsymbol{\gamma} := \left(\prod_{c=1}^C \frac{\mathbb{E}[Q_c] \cdot S - \mathbb{E}[Q_c^2]}{\mathbb{E}[Q_c^2] - \mathbb{E}[Q_c]^2}\right)^{1/C} \left(\frac{\mathbb{E}[\boldsymbol{Q}]}{\sum_{c=1}^C \mathbb{E}[Q_c]}\right) \tag{27}$$

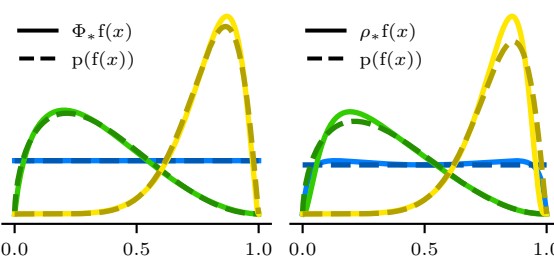

Figure 2: Approximating Gaussian pushforwards $\Phi_* f(x)$ and $\rho_* f(x)$ through normCDF and sigmoid respectively by Beta distributions $p(f(x))$ with moment matching. With norm-CDF, we can match exact moments, whereas with sigmoid, we match approximate moments. Moreover, when $f(x)$ is the standard Normal $\mathcal{N}(0,1)$ (on the left in blue), the approximation for the pushforward through normCDF is exact (Appendix D.1).

for $h(x)$, where $S = \max(\sum_{c=1}^{C} \mathbb{E}[Q_c], 1)$. Taking the maximum with $1$ in the expression for $S$ is an additional approximation to ensure $\boldsymbol{\gamma} > \mathbf{0}$. Note that by performing such moment matching to obtain the Dirichlet parameters in Eq. (27), the mean of this Dirichlet matches precisely the approximate predictive Eq. (12). That is, *the approximate predictive of the exact distributional model* (Eq. (11)) *matches the exact predictive of the approximate distributional model* (Eq. (20)).

## 4.2 Intermediate Beta Matching

When $\varphi = \Phi$ or $\rho$, noting that their range is $(0, 1)$, we can decompose the approximate distributional model (Eq. (20)) further:

$$h: \mathcal{X} \xrightarrow{\ f\ } \mathcal{G}(\mathbb{R}^C) \xrightarrow{\ p\ } \mathcal{B}((0,1))^C \xrightarrow{\ n\ } \mathcal{D}(\Delta^{C-1}), \tag{28}$$

where $\mathcal{B}((0,1))$ is the space of Beta distributions on $(0, 1)$. Like the output Dirichlets, we obtain the intermediate Beta distributions by moment matching (Appendix D). Figure 2 indicates that such Beta approximations tend to be of high quality. In Appendix D.1, we give an information-geometric motivation for this, which shows that the choice of activation $\Phi$ is ideal for approximating the Gaussian pushforwards with Beta distributions. The quality of such Beta approximations is encouraging for that of the output Dirichlet approximations. Analysing this theoretically presents an interesting direction for future research.

## 5 Training

With the choices $\varphi = \Phi$ or $\rho$, we could still train the underlying model with a cross-entropy (CE) loss, replacing the softmax by the activation $\boldsymbol{a} = \boldsymbol{n} \circ \boldsymbol{\Phi}$ or $\boldsymbol{n} \circ \boldsymbol{\rho}$. However, we can instead leverage the fact that $\Phi$ and $\rho$ have values in $(0, 1)$ to train using the binary cross-entropy (BCE) loss.

The BCE loss is a valid choice of loss as it is a (negative) strictly proper scoring rule (see Appendix I). Moreover, it has been observed to provide advantageous training dynamics compared to the CE loss [53, 28]. The choice of $\varphi = \Phi$ or $\rho$ not only fits our framework for closed-form predictives, but allows one to leverage such advantageous training dynamics (see experimental results in Section 7).

For example, in linearised Laplace and SNGP, a neural network $\boldsymbol{f} \colon \mathcal{X} \to \mathbb{R}^C$ is trained for the mean of f, while the covariance is obtained post-hoc. Thus, for these methods, we can train the mean function $\boldsymbol{f}$ with the BCE loss

$$\mathcal{L}((x_n, c_n)_{n=1}^N) = -\sum_{n=1}^{N} \sum_{c=1}^{C} \left( \delta_{c_n,c} \log(\varphi(f_c(x))) + (1 - \delta_{c_n,c}) \log(1 - \varphi(f_c(x))) \right). \tag{29}$$

In contrast to the previously mentioned methods, HET directly trains not only for the mean $\boldsymbol{f}$ but also for the covariance matrix $\Sigma$ of f through the predictives. Thus, in this case, we use the BCE with the pushforward means $\bar{Q}_c := \mathbb{E}_{Q_c \sim \varphi_* f_c(x)}[Q_c]$:

$$\mathcal{L}((x_n, c_n)_{n=1}^N) = -\sum_{n=1}^{N} \sum_{c=1}^{C} \left( \delta_{c_n,c} \log(\bar{Q}_c) + (1 - \delta_{c_n,c}) \log(1 - \bar{Q}_c) \right). \tag{30}$$

# 6 Computational Gains

The runtime cost of the MC predictive approximation is $\mathcal{O}(S \cdot C + C^3)$, and the memory cost is $\mathcal{O}(S \cdot C + C^2)$, where $S$ is the number of samples. The $C^2$ term in memory comes from the logit covariance matrix; the $C^3$ term in runtime comes from the need to Cholesky-decompose such a covariance matrix before sampling. For example, on a ResNet-50, sampling one thousand logit vectors from the logit-space Gaussian incurs approximately 7% overhead on the forward pass: see Appendix H for MC timing experiments. In embedded systems with much smaller models, this overhead is even larger. In contrast, the runtime and memory cost of calculating our predictives is $\mathcal{O}(C)$, which is negligible in comparison.

Indeed, the approximation Eq. (12) uses only the mean and *the diagonal* of the logit covariance to calculate the predictive. That is, the logit quantities required to compute a predictive scale as $\mathcal{O}(C)$ instead of $\mathcal{O}(C^2)$. This provides further opportunities for computational savings in the methods used to obtain the Gaussian distributions. E.g., in HET, the logit covariance is trained with both a diagonal term and a low-rank term. Our framework thus allows dropping the low-rank term (Appendix J.2).

# 7 Experiments

We now investigate our two research questions:

(i) Do we have to sacrifice performance for sample-free predictives? (Section 7.1)

(ii) What are the effects of changing the learning objective? (Section 7.2)

To answer the first question, we consider fixed (method, activation) pairs and verify that our methods perform on par with the Monte Carlo sampled predictives. In the main paper, we report results on **Softmax** ($\varphi = \exp$, trained with vanilla CE), **NormCDF** ($\varphi = \Phi$) and **Sigmoid** ($\varphi = \rho$) models. In Appendix K.1, we investigate the performance of Softmax models with closed-form predictives.

We consider **Heteroscedastic Classifiers (HET)** [7], **Spectral-Normalised Gaussian Processes (SNGP)** [31], and last-layer **Laplace-Approximated Networks** [11] as backbones supplying logit-space Gaussian distributions. The resulting 21 (method, activation, predictive) triplets are evaluated on ImageNet-1k [12], CIFAR-10, and CIFAR-100 [26] on five metrics aligning with practical needs from uncertainty estimates [39]. Further, we test the Out-of-Distribution (OOD) detection capabilities of the models on balanced mixtures of In-Distribution (ID) and OOD samples. For ImageNet, we treat ImageNet-C [18] samples with 15 corruption types and 5 severity levels as OOD samples. For CIFAR-10 and CIFAR-100, we use the CIFAR-10C corruptions.

Table 1: Comparison of ECE results for different predictives using a fixed Laplace backbone.

| Method | Mean $\pm$ 2 std |
|---|---|
| **Sigmoid Laplace** | |
| Closed-Form | $0.0101 \pm 0.0006$ |
| MC 1000 | $0.0153 \pm 0.0011$ |
| MC 100 | $0.0153 \pm 0.0012$ |
| MC 10 | $0.0165 \pm 0.0014$ |
| **NormCDF Laplace** | |
| MC 10 | $0.0112 \pm 0.0005$ |
| Closed-Form | $0.0114 \pm 0.0007$ |
| MC 100 | $0.0115 \pm 0.0006$ |
| MC 1000 | $0.0118 \pm 0.0007$ |

For the second question, we evaluate our closed-form predictives and moment-matched Dirichlets against softmax models with the Laplace Bridge [19] and Mean Field [32] approximations, the linear-time approximation by Shekhovtsov and Flach [48], and MC sampling; see Section 7.2.

To provide a fair comparison, we reimplement each method as simple-to-use wrappers around deterministic backbones. For ImageNet evaluation, we use a ResNet-50 backbone from the `timm` library [53] pretrained with the softmax activation function, and train each (method, activation) pair for 50 ImageNet-1k epochs following Mucsányi et al. [40]. For Vision Transformer [14] experiments, refer to Appendix K.2. On CIFAR-10, we train ResNet-28 models from scratch for 100 epochs. The CIFAR-100 experiments train WideResNet-28-5 models for 200 epochs. We search for ideal hyperparameters with a ten-step Bayesian Optimization scheme [47] in Weights & Biases [5].

The main paper focuses on ImageNet results to highlight the scalability of our framework and only shows proper scoring results for CIFAR-10. For a complete overview of CIFAR-10 and CIFAR-100 results, refer to Appendix K.3 and Appendix K.4, respectively.

In all plots of the main paper and the Appendix, the error bars are calculated over five independent runs with different seeds and show the minimum and maximum metric value.

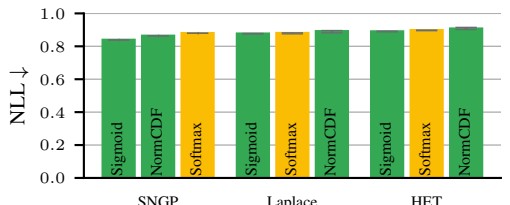

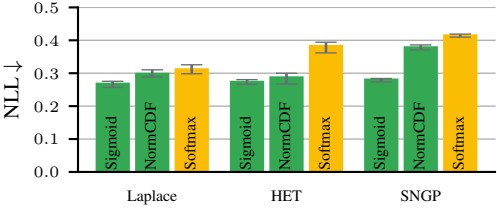

(a) Closed-form predictives (■) are either on par with or outperform Softmax (■) across all methods.

(b) Closed-form predictives (■) consistently outperform Softmax (■) across all methods.

Figure 3: ImageNet (left) and CIFAR-10 (right) test negative log likelihood results.

## 7.1 Quality of Sample-Free Predictives

In this section, we investigate whether there is a price to pay for sample-free predictives and second-order distributions. To this end, we take the two best-performing methods on the Expected Calibration Error (ECE) metric [42]—NormCDF and Sigmoid Laplace (see Fig. 4a)—and evaluate their per-predictive performance. As Table 1 shows, the closed-form predictive is always on par with the (unbiased) MC estimates on ImageNet while being cheaper. Appendix K.5 reports the KL divergence of different approximations to the 'true' predictive (estimated using 10,000 MC samples) on ImageNet. Table 2 shows that the moment-matched second-order Dirichlet distributions consistently outperform both MC and sample-free predictives on the OOD detection task, showcasing the practical utility of second-order distributional knowledge.

## 7.2 Effects of Changing the Learning Objective

The previous section shows that for models trained with Sigmoid or NormCDF, our closed-form predictives always perform on par with MC sampling

Table 2: Comparison of OOD AUROC results on ImageNet using severity level one for different uncertainty estimators on the three best-performing (method, activation) pairs: Sigmoid Laplace, NormCDF HET, and NormCDF Laplace. The best estimator is derived from the moment-matched Dirichlet distributions in all three cases. MC estimates use 1000 samples.

| Method | Mean $\pm$ 2 std |
|---|---|
| **Sigmoid Laplace** | |
| Dirichlet Mutual Information | $0.6388 \pm 0.0005$ |
| Closed-form Predictive Entropy | $0.6353 \pm 0.0003$ |
| Monte Carlo Expected Entropy | $0.6321 \pm 0.0001$ |
| **NormCDF HET** | |
| Dirichlet Expected Entropy | $0.6353 \pm 0.0021$ |
| Monte Carlo Expected Entropy | $0.6281 \pm 0.0020$ |
| Closed-form Predictive Entropy | $0.6277 \pm 0.0021$ |
| **NormCDF Laplace** | |
| Dirichlet Expected Entropy | $0.6321 \pm 0.0009$ |
| Monte Carlo Predictive Entropy | $0.6296 \pm 0.0012$ |
| Closed-Form Predictive Entropy | $0.6296 \pm 0.0012$ |

while being more efficient. However, our proposed objective (see Eq. (29)) changes the training dynamics and, subsequently, the performance of models. In this section, we showcase the performance of methods equipped with our closed-form predictives and learning objectives against Softmax models. For the latter, we use the *best-performing* predictive and estimator (see Appendices E and F for an overview of predictive approximations and estimators). We employ our methods with the closed-form predictives and second-order Dirichlet distributions to demonstrate their competitive nature while being more efficient than MC sampling.

We first evaluate how calibrated the models are using the test NLL [17] and Expected Calibration Error (ECE) [42] metrics. Fig. 3b shows that on CIFAR-10, the NLL score of our closed-form predictives (Sigmoid, NormCDF) is consistently better than the corresponding Softmax results for all methods. On the large-scale ImageNet dataset, Sigmoid and NormCDF predictives either outperform or are on par with Softmax (see Fig. 3a). The ECE metric requires the models' confidence (i.e., maximum class probability) to match their accuracy. Fig. 4a shows that on ImageNet, our closed-form predictives have a clear advantage over Softmax across all methods.

Next, we turn to the correctness prediction task: whether the models can predict the correctness of their own predictions. We consider *correctness estimators* $t(x) := \max_{1 \le c \le C} \mathbb{E}_{\mathbf{P} \sim \mathrm{h}(x)}[P_c] \in [0, 1]$ for inputs $x \in \mathcal{X}$ derived from the predictive mean $\mathbb{E}_{\mathbf{P} \sim \mathrm{h}(x)}[\mathbf{P}]$. Framed as a binary prediction task, the goal of these estimators is to predict the probability of the predicted class' correctness, defined as $v(x) = [\hat{y}(x) = y] \in \{0, 1\}$ with $\hat{y}(x) := \arg\max_{1 \le c \le C} \mathbb{E}_{\mathbf{P} \sim \mathrm{h}(x)}[P_c] \in \{1, \ldots, C\}$. We then

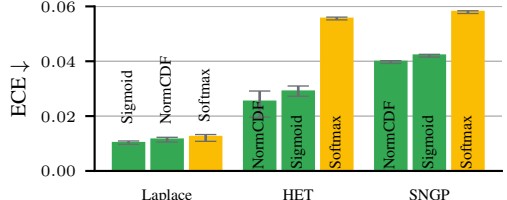

(a) Closed-form predictives (■) are more calibrated than Softmax (■) across all methods. Note the restricted $y$-limits for readability.

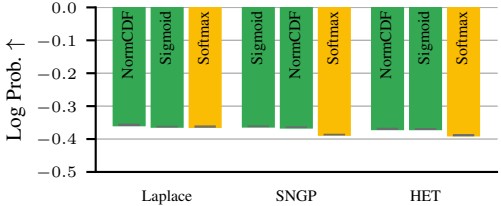

(b) Our closed-form predictives (■) consistently outperform Softmax (■) on predicting their own correctness.

Figure 4: ImageNet ECE (left) and log probability proper scoring results for the binary correctness prediction task (right).

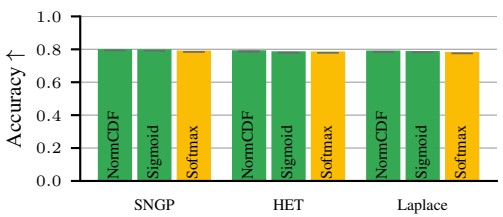

(a) **Closed-form predictives do not sacrifice accuracy.** Closed-form predictives (■) either outperform or are on par with Softmax (■) across all methods.

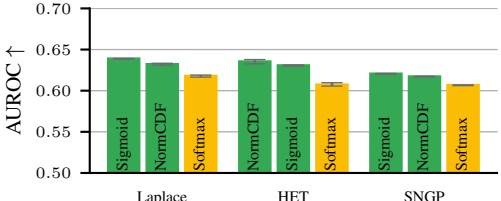

(b) **Closed-form predictives yield superior OOD detection performance.** Across all methods, the best-performing predictive is closed-form (■).

Figure 5: ImageNet validation accuracy (left) and OOD detection AUROC results for ImageNet-C transforms of severity level one (right).

measure the binary log probability score of $\tilde{t}(x)$, given by $v(x) \log t(x) + (1 - v(x)) \log(1 - t(x))$. Fig. 4b shows that our closed-form predictives outperform all Softmax predictives across all methods.

**Closed-form predictives do not sacrifice accuracy.** Fig. 5a evidences this claim on ImageNet: our closed-form predictives either outperform or are on par with Softmax predictives. These results are in line with the findings of Wightman et al. [54], who also recommend training with a per-class binary cross-entropy loss using the sigmoid activation function. Appendix K.6 shows that the BCE loss already yields performance gains in the model backbones (without the logit-space Gaussians).

Finally, we consider another binary prediction task, where a general uncertainty estimator $u(x) \in \mathbb{R}$ (derived from predictives or second-order Dirichlet distributions) is tasked to separate ID and OOD samples from a balanced mixture thereof. As the uncertainty estimator can take on any real value, we measure the Area Under the Receiver Operating Characteristic curve (AUROC), which quantifies the separability of ID and OOD samples w.r.t. the uncertainty estimator. As OOD inputs, we consider corrupted ImageNet-C [18] samples. Fig. 5b shows that our closed-form predictives outperform Softmax across all methods. Importantly, these closed-form predictives and second-order Dirichlet distributions are considerably cheaper to calculate than Softmax MC predictions (see Section 6). See Appendix K.7 for the other severity levels.

## 8 Conclusion and Limitations

We developed a framework that allows for obtaining predictives and other quantities of interest from logit space Gaussian distributions in closed form. Our experimental results suggest that the ubiquitous softmax activation should be replaced by normCDF or sigmoid for uncertainty quantification tasks.

A limitation is that our approximate predictives and approximate Dirichlet distributions do not encode correlations between classes. While our theoretical analysis suggests strong bounds on the approximations, leveraging more expressive yet tractable second-order distributions on the simplex that can model correlations presents an interesting area of future research.

## Acknowledgments

The authors gratefully acknowledge co-funding by the European Union (ERC, ANUBIS, 101123955). Views and opinions expressed are however those of the author(s) only and do not necessarily reflect those of the European Union or the European Research Council. Neither the European Union nor the granting authority can be held responsible for them. BM & PH are supported by the DFG through Project HE 7114/6-1 in SPP2298/2. NDC is supported by the Fonds National de la Recherche, Luxembourg, Project 17917615. PH is a member of the Machine Learning Cluster of Excellence, funded by the Deutsche Forschungsgemeinschaft (DFG, German Research Foundation) under Germany's Excellence Strategy – EXC number 2064/1 – Project number 390727645. The authors also gratefully acknowledge the German Federal Ministry of Education and Research (BMBF) through the Tübingen AI Center (FKZ: 01IS18039A); and funds from the Ministry of Science, Research and Arts of the State of Baden-Württemberg.

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

# A Gaussian Integral Derivations

In this appendix, we derive the closed-form formula for the mean of Gaussian pushforwards through exp (Eq. (8)) and normCDF (Eq. (13)), as well as the approximations for pushforwards through sigmoid (Eq. (15)) and softmax.

## A.1 Gaussian Exp Integral

By absorbing the exponential into the Gaussian probability density function, we get

$$
\begin{aligned}
\int_{\mathbb{R}} \exp(y) \, \mathcal{N}(\mu, \sigma^2)(dy) &= \int_{\mathbb{R}} \exp(y) \exp\left(-\frac{(y-\mu)^2}{2\sigma^2}\right) \, dy \\
&= \int_{\mathbb{R}} \exp\left(\mu + \frac{\sigma^2}{2}\right) \exp\left(-\frac{1}{2\sigma^2}(y - \mu - \sigma^2)^2\right) \, dy \\
&= \int_{\mathbb{R}} \exp\left(\mu + \frac{\sigma^2}{2}\right) \mathcal{N}(\mu + \sigma^2, \sigma^2)(dy) \\
&= \exp\left(\mu + \frac{\sigma^2}{2}\right).
\end{aligned}
\tag{31}
$$

## A.2 Gaussian NormCDF Integral

Here, we derive the classical normCDF Gaussian integration formula [43, Eq. 10,010.8].

For $\lambda > 0$, $Z \sim \mathcal{N}(0,1)$ and $Y \sim \mathcal{N}(\mu, \sigma^2)$ with $Y$ and $Z$ independent,

$$
\begin{aligned}
\int_{\mathbb{R}} \Phi(\lambda y) \, \mathcal{N}(\mu, \sigma^2)(dy) &= \int_{\mathbb{R}} p(Z \leq y/\lambda) \, \mathcal{N}(\mu, \sigma^2)(dy) \\
&= p\left(Z \leq \frac{Y}{\lambda}\right) \\
&= p\left(\frac{Z/\lambda - Y + \mu}{\sqrt{\lambda^{-2} + \sigma^2}} \leq \frac{\mu}{\sqrt{\lambda^{-2} + \sigma^2}}\right) \\
&= \Phi\left(\frac{\mu}{\sqrt{\lambda^{-2} + \sigma^2}}\right)
\end{aligned}
\tag{32}
$$

where we used $\frac{Z/\lambda - X + \mu}{\sqrt{\lambda^{-2} + \sigma^2}} \sim \mathcal{N}(0,1)$ for the last equality. Taking $\lambda = 1$, this gives the formula for the exact predictive with normCDF.

## A.3 Probit Approximation for Gaussian Sigmoid Integral

Taylor expanding $\rho$ to first order about 0,

$$
\begin{aligned}
\rho(y) &= \frac{1}{2} + \frac{1}{4}y + o(y), \\
\Phi(y) &= \frac{1}{2} + \frac{1}{\sqrt{2\pi}}y + o(y).
\end{aligned}
\tag{33}
$$

as $y \to 0$. Hence matching $\rho$ and $\Phi$ to first order we get the approximation $\rho(y) \approx \Phi\left(\sqrt{\frac{\pi}{8}}y\right)$. So using Eq. (32) we derive the *probit approximation* [49, 34]

$$
\int_{\mathbb{R}} \rho(y) \, \mathcal{N}(\mu, \sigma^2)(dy) \overset{(1)}{\approx} \int_{\mathbb{R}} \Phi\left(\sqrt{\frac{\pi}{8}}y\right) \mathcal{N}(\mu, \sigma^2)(dy) = \Phi\left(\frac{\mu}{\sqrt{\frac{8}{\pi} + \sigma^2}}\right) \overset{(2)}{\approx} \rho\left(\frac{\mu}{\sqrt{1 + \frac{\pi}{8}\sigma^2}}\right).
\tag{34}
$$

Note that the approximation (2) is not strictly needed, as $\Phi$ is computationally tractable. However, adding (2) empirically improves the quality of the overall approximation. This may be due to the fact the thicker tails of $\rho$ in the integrand of the left-hand side are better captured by $\rho$ than $\Phi$ on the right-hand side.

### A.4 Mean Field Approximation for Gaussian Softmax Integral

For $\boldsymbol{\mu} \in \mathbb{R}^C$, $\boldsymbol{\sigma}^2 \in \mathbb{R}^C_{>0}$ and $\boldsymbol{\Sigma} = \mathbf{diag}(\boldsymbol{\sigma}^2)$, the *mean field approximation* to the Gaussian softmax integral [32] is obtained as follows

$$
\begin{aligned}
\mathbb{E}_{\boldsymbol{Y} \sim \mathcal{N}(\boldsymbol{\mu}, \boldsymbol{\Sigma})}[\mathrm{softmax}_c \, \boldsymbol{Y}] &= \mathbb{E}\left[\left(2 - C + \sum_{c' \neq c} \rho(Y_c - Y_{c'})^{-1}\right)^{-1}\right] \\
&\stackrel{(1)}{\approx} \left(2 - C + \sum_{c' \neq c} \mathbb{E}[\rho(Y_c - Y_{c'})]^{-1}\right)^{-1} \\
&\stackrel{(2)}{\approx} \left(2 - C + \sum_{c' \neq c} \mathbb{E}[\rho(Y_c - \mu_{c'})]^{-1}\right)^{-1} \\
&\stackrel{(3)}{\approx} \left(2 - C + \sum_{c' \neq c} \rho\left(\frac{\mu_c - \mu_{c'}}{\sqrt{1 + \frac{\pi}{8}\sigma_c^2}}\right)^{-1}\right)^{-1} \\
&= \mathrm{softmax}_c\left(\frac{\boldsymbol{\mu}}{\sqrt{1 + \frac{\pi}{8}\boldsymbol{\sigma}^2}}\right)
\end{aligned}
\tag{35}
$$

i.e.,

$$
\mathbb{E}_{\boldsymbol{Y} \sim \mathcal{N}(\boldsymbol{\mu}, \boldsymbol{\Sigma})}[\mathbf{softmax}\, \boldsymbol{Y}] \approx \mathbf{softmax}\left(\frac{\boldsymbol{\mu}}{\sqrt{1 + \frac{\pi}{8}\boldsymbol{\sigma}^2}}\right).
\tag{36}
$$

(1) is the mean field approximation, and (3) uses the probit approximation Eq. (34). [32] provides two other variants of this approximation with other choices of approximation (2).

## B Comparison with the Multinomial Probit Model

In this appendix, we show that a model whose output activation is a composition of an element-wise normCDF activation $\boldsymbol{\Phi}$ and a normalisation $\boldsymbol{n}$ is distinct from the classical multinomial probit model [9].

Given a logit $\boldsymbol{y} \in \mathbb{R}^C$, the multinomial probit model sets

$$
Z(\boldsymbol{y}) = \underset{1 \leq c \leq C}{\arg\max}\, Y_c
\tag{37}
$$

where $Y_c = y_c + \epsilon_c$ and the $\epsilon_c$ are i.i.d. standard Normal. So

$$
p(c \mid y) = p(y_c + \epsilon_c > y_{c'} + \epsilon_{c'} \; \forall\, c' \neq c)
\tag{38}
$$

which is generally not analytically tractable. On the other hand, a model that uses normCDF and normalisation as output activation yields

$$
p(c \mid y) = \frac{p(y_c + \epsilon_c > 0)}{\sum_{c'=1}^{C} p(y_{c'} + \epsilon_{c'} > 0)} = \frac{\Phi(y_c)}{\sum_{c'=1}^{C} \Phi(y_c')}.
\tag{39}
$$

In the case $C = 2$, the multinomial probit model Eq. (38) outputs closed-form probabilities. This allows us to construct an explicit counterexample to the equivalence of the two models Eq. (38) and Eq. (39):

$$
p(y_1 + \epsilon_1 > y_2 + \epsilon_2) = p\left(\frac{y_1 - y_2}{2} + \frac{\epsilon_1 - \epsilon_2}{2} > 0\right) = \Phi\left(\frac{y_1 - y_2}{2}\right) \neq \frac{\Phi(y_1)}{\Phi(y_1) + \Phi(y_2)}.
\tag{40}
$$

## C Theoretical Analyses

In this appendix, we provide (formal and informal) theoretical analyses of the quality of various predictive approximations. This complements the empirical analyses, for instance in the synthetic experiment (Fig. 1) or in [10].

Due to its information-theoretic interpretation, a natural divergence to equip the probability simplex $\Delta^{C-1}$ with is the Kullback-Leibler (KL) divergence

$$D_{KL}(\boldsymbol{p}, \boldsymbol{q}) = \sum_{c=1}^{C} p_c(\log p_c - \log q_c) \tag{41}$$

which is well defined if $\boldsymbol{p}$, $\boldsymbol{q}$ lie in the interior of the simplex ($p_i, q_i \neq 0, 1$ for all $i$). So, for a predictive approximation, $\hat{\boldsymbol{p}}$, we would like to analyse $D_{KL}(\boldsymbol{p}, \hat{\boldsymbol{p}})$, where $\boldsymbol{p} := \mathbb{E}_{\boldsymbol{P} \sim \boldsymbol{a}_* \mathcal{N}(\boldsymbol{\mu}, \boldsymbol{\Sigma})}[\boldsymbol{P}]$ is the true predictive, $\boldsymbol{\mu}$ and $\boldsymbol{\Sigma}$ are some logit space mean and covariance and $\boldsymbol{a}$ is an output activation (e.g. $\boldsymbol{a} = \boldsymbol{n} \circ \boldsymbol{\varphi}$).

### C.1 Informal Analysis of Monte Carlo Approximations

An $N$ sample Monte Carlo estimate is defined as

$$\hat{\boldsymbol{P}}^S := \frac{1}{S} \sum_{s=1}^{S} \hat{\boldsymbol{P}}^{(s)} \tag{42}$$

where the $\hat{\boldsymbol{P}}^{(s)}$ are i.i.d. $\boldsymbol{a}_* \mathcal{N}(\boldsymbol{\mu}, \boldsymbol{\Sigma})$. The computational cost of MC integration is $\mathcal{O}(S \cdot C)$. This becomes prohibitive for large $S$ and $C$. Thus, for a fair assessment of the quality of such an estimate in terms of the number of classes, one should consider MC estimates $\hat{\boldsymbol{P}}^{\lceil S/C \rceil}$.

We now give an informal theoretical argument for the linear growth of the KL divergence between $\boldsymbol{p}$ and $\hat{\boldsymbol{P}}^{\lceil S/C \rceil}$ in terms of $C$, under the distributional conditions of the synthetic experiment (Fig. 1).

Taylor expanding Eq. (41) about $\boldsymbol{p}$ to second order we obtain

$$\begin{aligned}
D_{KL}(\boldsymbol{p}, \boldsymbol{q}) &= \sum_{c=1}^{C} p_c(\log p_c - \log q_c) \\
&\stackrel{(1)}{\approx} \sum_{c=1}^{C} p_c \left( 1 - \frac{p_c}{q_c} + \frac{(p_c - q_c)^2}{2p_c^2} \right) \\
&= \underbrace{\sum_{c=1}^{C} p_c}_{=1} - \underbrace{\sum_{c=1}^{C} q_c}_{=1} + \sum_{c=1}^{C} \frac{(p_c - q_c)^2}{2p_c} \\
&= \sum_{c=1}^{C} \frac{(p_c - q_c)^2}{2p_c} \\
&\stackrel{(2)}{\approx} \frac{C}{2} \|\boldsymbol{p} - \boldsymbol{q}\|_2^2
\end{aligned} \tag{43}$$

where approximation (1) assumes $\|\boldsymbol{p} - \boldsymbol{q}\|_2$ is small, and (2) assumes $p_c \approx 1/C$.

In the synthetic experiment (Fig. 1), the logit class-wise means $\mu_c$ and variances $\sigma_c^2$ are sampled in an i.i.d. way. Let $\boldsymbol{Q} \sim \mathcal{N}(\boldsymbol{\mu}, \mathbf{diag}(\boldsymbol{\sigma}^2))$ the unnormalised 'probabilities' and $\boldsymbol{P} := \boldsymbol{Q}/\sum_{c=1}^{C} Q_c$ the probabilities. We have

$$\mathrm{Var}[\boldsymbol{P}] = \mathbb{E}\left[ \frac{\boldsymbol{Q}^2}{\left(\sum_{c=1}^{C} Q_c\right)^2} \right] - \mathbb{E}\left[ \frac{\boldsymbol{Q}}{\sum_{c=1}^{C} Q_c} \right]^2 \approx \frac{\mathbb{E}[\boldsymbol{Q}^2] - \mathbb{E}[\boldsymbol{Q}]^2}{\left(\sum_{c=1}^{C} \mathbb{E}[Q_c]\right)^2} \approx \frac{\mathrm{Var}[\boldsymbol{Q}]}{C^2 \mathbb{E}[Q_1]^2} \tag{44}$$

where all operations are taken element-wise, and $\mathbb{E}[Q_c] \approx \mathbb{E}[Q_1]$ follows from the fact that the $\mu_c$ and $\sigma_c^2$ are i.i.d. Thus

$$\mathbb{E}\left[\|\boldsymbol{p} - \boldsymbol{P}\|_2^2\right] = \sum_{c=1}^{C} \mathrm{Var}[P_c] \approx \sum_{c=1}^{C} \frac{\mathrm{Var}[Q_c]}{C^2 \mathbb{E}[Q_1]^2} \approx \frac{\mathrm{Var}[Q_1]}{C \mathbb{E}[Q_1]^2}. \tag{45}$$

Now the MC samples $\hat{\boldsymbol{P}}^{(s)}$ are i.i.d. copies of $\boldsymbol{P}$. So we have

$$\mathbb{E}\left[\|\boldsymbol{p} - \hat{\boldsymbol{P}}^{\lceil S/C \rceil}\|_2^2\right] = \frac{1}{\lceil S/C \rceil} \mathbb{E}\left[\|\boldsymbol{p} - \boldsymbol{P}\|_2^2\right] \approx \mathbb{E}\left[\frac{\mathrm{Var}[Q_1]}{S \mathbb{E}[Q_1]^2}\right]. \tag{46}$$

where the outer expectation is over the (third-order) distributions over $\boldsymbol{P}$. Plugging this into Eq. (43) we get

$$\mathbb{E}[\mathrm{D_{KL}}(\boldsymbol{p}, \hat{\boldsymbol{P}}^{\lceil S/C \rceil})] \approx \mathbb{E}\left[\frac{\mathrm{Var}[Q_1]}{2\mathbb{E}[Q_1]^2}\right] \cdot \frac{C}{S} \tag{47}$$

which grows linearly with the number of classes, as observed in Fig. 1.

## C.2 Analysis of the Closed-Form Approximations (Theorem 3.1)

In this section we extend and prove Theorem 3.1, and discuss whether its underlying assumptions are fulfilled in practice. We start with a lemma.

**Lemma C.1.** *Let $\mathcal{Y}$ be a family of $(0, M)$-valued random variables such that $\sup_{Y \in \mathcal{Y}} \mathbb{E}[Y^{-k}] < \infty$ for all $k > 0$. Then for $X, Y$ $(0, M)$-valued random variables with $Y \in \mathcal{Y}$ and any $\epsilon > 0$,*

$$\left|\mathbb{E}\left[\frac{X}{Y}\right] - \frac{\mathbb{E}[X]}{\mathbb{E}[Y]}\right| = O\left(\mathrm{Var}(X)^{\frac{1}{2}-\epsilon} \mathrm{Var}(Y)^{\frac{1}{2}} + \mathrm{Var}(Y)^{1-\epsilon}\right) \tag{48}$$

*as $\mathrm{Var}(Y) \to 0$ or $\mathrm{Var}(X), \mathrm{Var}(Y) \to 0$.*

*Proof.* Let $X, Y$ be $(0, M)$-valued random variables. We take a bivariate Taylor expansion of $\frac{X}{Y}$ for the random variables $X$ and $Y$ around $\mathbb{E}[X]$ and $\mathbb{E}[Y]$ respectively:

$$\begin{aligned}
\frac{X}{Y} =& \frac{\mathbb{E}[X]}{\mathbb{E}[Y]} + \frac{1}{\mathbb{E}[Y]}(X - \mathbb{E}[X]) - \frac{\mathbb{E}[X]}{\mathbb{E}[Y]^2}(Y - \mathbb{E}[Y]) \\
& - \frac{1}{\xi(Y)^2}(X - \mathbb{E}[X])(Y - \mathbb{E}[Y]) + \frac{\eta(X)}{\xi(Y)^3}(Y - \mathbb{E}[Y])^2
\end{aligned} \tag{49}$$

where $\eta(X) \in [\mathbb{E}[X], X]$ or $[X, \mathbb{E}[X]]$ and $\xi(Y) \in [\mathbb{E}[Y], Y]$ or $[Y, \mathbb{E}[Y]]$, using the Lagrange form of the remainder. So by the multivariate Hölder inequality we have for any $0 < \epsilon < 1/2$,

$$\begin{aligned}
\left|\mathbb{E}\left[\frac{X}{Y}\right] - \frac{\mathbb{E}[X]}{\mathbb{E}[Y]}\right| =& \left|-\mathbb{E}\left[\frac{1}{\xi(Y)^2}(X - \mathbb{E}[X])(Y - \mathbb{E}[Y])\right] + \mathbb{E}\left[\frac{\eta(X)}{\xi(Y)^3}(Y - \mathbb{E}[Y])^2\right]\right| \\
\leq& \mathbb{E}\left[\frac{1}{\xi(Y)^{2/\epsilon}}\right]^{\epsilon} \mathbb{E}[(X - \mathbb{E}[X])^{\frac{2}{1-2\epsilon}}]^{\frac{1}{2}-\epsilon} \mathbb{E}[(Y - \mathbb{E}[Y])^2]^{\frac{1}{2}} \\
& + \mathbb{E}\left[\frac{\eta(X)^{1/\epsilon}}{\xi(Y)^{3/\epsilon}}\right]^{\epsilon} \mathbb{E}[(Y - \mathbb{E}[Y])^{\frac{2}{1-\epsilon}}]^{1-\epsilon}.
\end{aligned} \tag{50}$$

Note that

$$\mathbb{E}[(X - \mathbb{E}[X])^{\frac{2}{1-2\epsilon}}] \leq M^{\frac{4\epsilon}{1-2\epsilon}} \mathbb{E}[(X - \mathbb{E}[X])^2] = M^{\frac{4\epsilon}{1-2\epsilon}} \mathrm{Var}(X),$$

$$\eta(X)^{1/\epsilon} \leq M^{1/\epsilon},$$

$$\sup_{Y \in \mathcal{Y}} \mathbb{E}\left[\frac{1}{\xi(Y)^k}\right] \leq \sup_{Y \in \mathcal{Y}} \mathbb{E}\left[\frac{1}{\min(Y, \mathbb{E}[Y]^k)}\right] \leq \sup_{Y \in \mathcal{Y}} \mathbb{E}\left[\frac{1}{Y^k} + \frac{1}{\mathbb{E}[Y]^k}\right] < \infty \text{ for } k \in \{2/\epsilon, 3/\epsilon\}. \tag{51}$$

So from Eq. (50) we obtain

$$\left|\mathbb{E}\left[\frac{X}{Y}\right] - \frac{\mathbb{E}[X]}{\mathbb{E}[Y]}\right| = O\left(\mathrm{Var}(X)^{\frac{1}{2}-\epsilon} \mathrm{Var}(Y)^{\frac{1}{2}} + \mathrm{Var}(Y)^{1-\epsilon}\right) \tag{52}$$

as $\mathrm{Var}(Y) \to 0$ or $\mathrm{Var}(X), \mathrm{Var}(Y) \to 0$. $\square$

Now recall our notation

$$q = q(\mu, \sigma^2) := \mathbb{E}_{Q \sim \rho_* \mathcal{N}(\mu, \sigma^2)}[Q] \approx \rho\left(\frac{\mu}{\sqrt{1 + \frac{\pi}{8}\sigma^2}}\right) =: \hat{q}(\mu, \sigma^2) = \hat{q} \tag{53}$$

where $\hat{q}$ is the probit approximation (Appendix A.3). Moreover we write $q_c := q(\mu_c, \sigma_c^2)$, $\hat{q}_c := \hat{q}(\mu_c, \sigma_c^2)$, $\boldsymbol{p}$ for the true predictive, $\hat{\boldsymbol{p}}$ for our approximate predictive, and in addition $\tilde{\boldsymbol{p}}$ for the approximate predictive using the exact one dimensional integrals, i.e.

$$\boldsymbol{p} := \mathbb{E}_{\boldsymbol{P} \sim \boldsymbol{a}_* \mathcal{N}(\boldsymbol{\mu}, \boldsymbol{\Sigma})}[\boldsymbol{P}], \quad \hat{\boldsymbol{p}} := \frac{\hat{q}_c}{\sum_{c'=1}^C \hat{q}_{c'}}, \quad \tilde{\boldsymbol{p}} := \frac{q_c}{\sum_{c'=1}^C q_{c'}}, \tag{54}$$

where $\boldsymbol{\Sigma}$ has $\boldsymbol{\sigma}$ as diagonal. We now restate Theorem 3.1 with an explicit expression for $M(\mathcal{K})$ and an additional decay result for the $O$ term:

**Theorem C.2.** *Suppose the means and variances $(\mu_c, \sigma_c^2)$ lie in some compact set $\mathcal{K} \subset \mathbb{R} \times [0, \infty)$ for each class c. Using the compactness of $\mathcal{K}$, define*

- $\delta(\mathcal{K}) := \sup_{(\mu, \sigma^2) \in \mathcal{K}}(\hat{q} - q)$,

- $u(\mathcal{K}) := \inf_{(\mu, \sigma^2) \in \mathcal{K}} q > 0$,

- $\Delta(\mathcal{K}) := \sup_{(\mu, \sigma^2) \in \mathcal{K}} \frac{q - \hat{q}}{q}$.

*If $\Delta(\mathcal{K}) > 1$ then*

$$D_{\mathrm{KL}}(\boldsymbol{p}, \hat{\boldsymbol{p}}) \leq \log\left(\frac{1 + \delta/u}{1 - \Delta}\right) + O \tag{55}$$

*where*

$$O = O\left(\mathrm{Var}\left(\sum_{c=1}^C Q_c\right)\right) \qquad as \ \mathrm{Var}\left(\sum_{c=1}^C Q_c\right) \to 0 \tag{56}$$

*and*

$$O = O\left(\max_{1 \leq c \leq C} \mathrm{Var}(Q_c)^{2-\epsilon}\right) \ \forall \epsilon > 0 \qquad as \ \mathrm{Var}(Q_c) \to 0 \ \forall 1 \leq c \leq C. \tag{57}$$

*Proof.* We have

$$D_{\mathrm{KL}}(\boldsymbol{p}, \hat{\boldsymbol{p}}) = \sum_{c=1}^C p_c(\log p_c - \log \hat{p}_c) = \underbrace{\sum_{c=1}^C p_c(\log p_c - \log \tilde{p}_c)}_{(1)} + \underbrace{\sum_{c=1}^C p_c(\log \tilde{p}_c - \log \hat{p}_c)}_{(2)}. \tag{58}$$

Assuming $\Delta < 1$, we can bound (2):

$$
\begin{aligned}
(2) &= \sum_{c=1}^C \frac{q_c}{\sum_{c'=1}^C q_{c'}}\left(\log\left(\frac{q_c}{\sum_{c'=1}^C q_{c'}}\right) - \log\left(\frac{\hat{q}_c}{\sum_{c'=1}^C \hat{q}_{c'}}\right)\right) \\
&= \frac{1}{\sum_{c=1}^C q_c}\sum_{c=1}^C q_c\left(-\log\left(\frac{\hat{q}_c}{q_c}\right) + \log\left(\frac{\sum_{c'=1}^C \hat{q}_{c'}}{\sum_{c'=1}^C q_{c'}}\right)\right) \\
&= \frac{1}{\sum_{c=1}^C q_c}\sum_{c=1}^C q_c\left(-\log\left(1 - \frac{q_c - \hat{q}_c}{q_c}\right) + \log\left(1 + \frac{\sum_{c'=1}^C \hat{q}_{c'} - \sum_{c'=1}^C q_{c'}}{\sum_{c'=1}^C q_{c'}}\right)\right) \\
&\leq \frac{1}{\sum_{c=1}^C q_c}\sum_{c=1}^C q_c\left(-\log(1 - \Delta) + \log\left(1 + \frac{C\delta}{Cu}\right)\right) \\
&= \frac{1}{\sum_{c=1}^C q_c}\sum_{c=1}^C q_c \log\left(\frac{1 + \delta/u}{1 - \Delta}\right) \\
&= \log\left(\frac{1 + \delta/u}{1 - \Delta}\right).
\end{aligned}
\tag{59}
$$

Now for (1), first note that $\mathbb{E}\left[\left(\sum_{c=1}^{C} Q_c\right)^{-k}\right] < \infty$ where[3] $Q_c \sim \rho_* \mathcal{N}(\mu_c, \sigma_c^2)$ and $k > 0$. By compactness of $\mathcal{K}$, we have in fact $\sup_{(\boldsymbol{\mu}, \boldsymbol{\sigma}^2) \in \mathcal{K}^C} \mathbb{E}\left[\left(\sum_{c=1}^{C} Q_c\right)^{-k}\right] < \infty$ for any $k > 0$. Thus we can apply Theorem C.1 with $\mathcal{Y} = \left\{\sum_{c=1}^{C} Q_c : (\boldsymbol{\mu}, \boldsymbol{\sigma}^2) \in \mathcal{K}^C\right\}$ to get that for any $\epsilon > 0$,

$$|p_c - \tilde{p}_c| = O\left(\mathrm{Var}(Q_c)^{\frac{1}{2}-\epsilon}\mathrm{Var}\left(\sum_{c'=1}^{C} Q_{c'}\right)^{\frac{1}{2}} + \mathrm{Var}\left(\sum_{c'=1}^{C} Q_{c'}\right)^{1-\epsilon}\right) \tag{60}$$

as $\mathrm{Var}\left(\sum_{c'=1}^{C} Q_{c'}\right) \to 0$ or $\mathrm{Var}(Q_c), \mathrm{Var}\left(\sum_{c'=1}^{C} Q_{c'}\right) \to 0$. Thus Taylor expanding each term of (1) around $p_c$ we get

$$(1) = \sum_{c=1}^{C} p_c \left(\frac{p_c - \tilde{p}_c}{p_c} + \frac{(p_c - \tilde{p}_c)^2}{2\omega_c(\tilde{p}_c)^2}\right)$$

$$= \underbrace{\sum_{c=1}^{C} p_c}_{=1} - \underbrace{\sum_{c=1}^{C} \tilde{p}_c}_{=1} + \sum_{c=1}^{C} p_c \frac{(p_c - \tilde{p}_c)^2}{2\omega_c(\tilde{p}_c)^2}$$

$$= \sum_{c=1}^{C} \frac{p_c}{2\omega_c(\tilde{p}_c)^2} \cdot O\left(\mathrm{Var}(Q_c)^{\frac{1}{2}-\epsilon}\mathrm{Var}\left(\sum_{c'=1}^{C} Q_{c'}\right)^{\frac{1}{2}} + \mathrm{Var}\left(\sum_{c'=1}^{C} Q_{c'}\right)^{1-\epsilon}\right)^2$$

$$= \sum_{c=1}^{C} \frac{p_c}{2\omega_c(\tilde{p}_c)^2} \cdot O\left(\mathrm{Var}(Q_c)^{1-2\epsilon}\mathrm{Var}\left(\sum_{c'=1}^{C} Q_{c'}\right)\right. \tag{61}$$

$$\left. + \mathrm{Var}(Q_c)^{\frac{1}{2}-\epsilon}\mathrm{Var}\left(\sum_{c'=1}^{C} Q_{c'}\right)^{\frac{3}{2}-\epsilon} + \mathrm{Var}\left(\sum_{c'=1}^{C} Q_{c'}\right)^{2-2\epsilon}\right)$$

$$= O\left(\sum_{c=1}^{C} \mathrm{Var}(Q_c)^{1-2\epsilon}\mathrm{Var}\left(\sum_{c'=1}^{C} Q_{c'}\right)\right.$$

$$\left. + \sum_{c=1}^{C} \mathrm{Var}(Q_c)^{\frac{1}{2}-\epsilon}\mathrm{Var}\left(\sum_{c'=1}^{C} Q_{c'}\right)^{\frac{3}{2}-\epsilon} + \mathrm{Var}\left(\sum_{c'=1}^{C} Q_{c'}\right)^{2-2\epsilon}\right)$$

where $\omega_c(\tilde{p}_c) \in [p_c, \tilde{p}_c]$ or $[\tilde{p}_c, p_c]$, where we used that $p_c$, $\tilde{p}_c$, and hence $\omega_c(\tilde{p}_c)$ is bounded away from 0 by compactness of $\mathcal{K}$. We now distinguish two cases. If we only assume $\mathrm{Var}\left(\sum_{c=1}^{C} Q_c\right) \to 0$, then we see from Eq. (61) that

$$(1) = O\left(\mathrm{Var}\left(\sum_{c=1}^{C} Q_c\right)\right). \tag{62}$$

If we make the stronger assumption $\mathrm{Var}(Q_c) \to 0$ for all $c$, then using the inequality

$$\mathrm{Var}\left(\sum_{c=1}^{C} Q_c\right) = \sum_{1 \leq c_1, c_2 \leq C} \mathrm{Cov}(Q_{c_1}, Q_{c_2}) \leq \sum_{1 \leq c_1, c_2 \leq C} \mathrm{Var}(Q_{c_1})^{1/2}\mathrm{Var}(Q_{c_2})^{1/2} \tag{63}$$

we see that $\mathrm{Var}\left(\sum_{c=1}^{C} Q_c\right) = O\left(\max_{1 \leq c \leq C} \mathrm{Var}(Q_c)\right)$ and hence, from Eq. (61),

$$(1) = O\left(\max_{1 \leq c \leq C} \mathrm{Var}(Q_c)^{2-2\epsilon}\right) \tag{64}$$

and the result follows since $\epsilon$ is arbitrary. $\qquad\square$

---

[3]In the case $\varphi = \Phi$ and $Q_c \sim \Phi_* \mathcal{N}(\mu_c, \sigma_c^2)$, due to the fast decay of the tail of $\Phi$ we may have $\mathbb{E}\left[\left(\sum_{c=1}^{C} Q_c\right)^{-k}\right] = \infty$, and the proof strategy fails in that case.

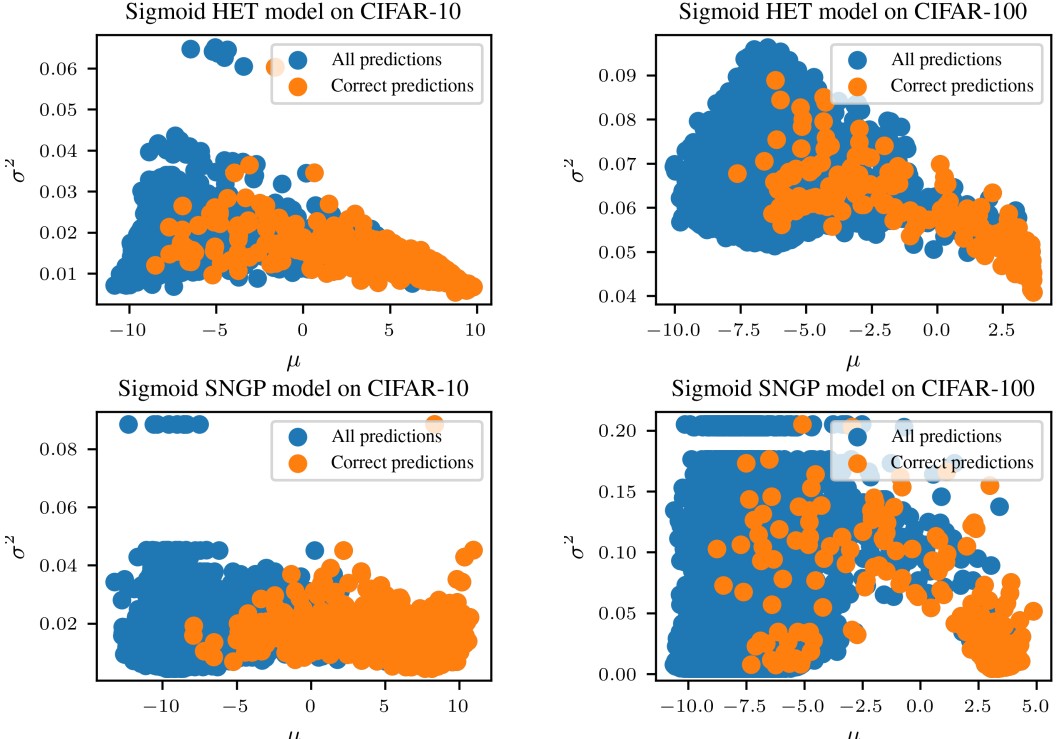

Figure C.1: Scatter plots of logit mean and variance pairs $(\mu_c, \sigma_c^2)$ on real models. We see that these seem to remain constrained to a compact set, with no much difference in the support as we increase the number of classes by an order of magnitude (from CIFAR-10 to CIFAR-100).

Equation (63) shows that requiring $\mathrm{Var}\left(\sum_{c=1}^{C} Q_c\right) \to 0$ is weaker than requiring $\mathrm{Var}(Q_c) \to 0$ for all $c$; but we see in Theorem C.2 that assuming the latter gives almost twice as strong a decay rate in $\mathrm{D}_{\mathrm{KL}}$. This means that neither Eq. (56) nor Eq. (57) can be proven from the other; both results are thus of interest.

A key assumption in Theorem C.2 is the logit means and variances being restricted to a compact set. In Fig. C.1 we see empirically that this is approximately the case for HET and SNGP models, independently of the number of classes.

Theorem C.2 is of practical value as $M(\mathcal{K}) := \log\left(\frac{1+\delta/u}{1-\Delta}\right)$ is well-behaved:

1. $M(\mathcal{K})$ is independent of $C$,

2. $M(\mathcal{K}) \to 0$ as $\delta \to 0$.

In other words, given knowledge of the worst case error in the approximation Eq. (15) on the compact set $\mathcal{K}$, we can bound the KL divergence in terms of that error independently of the number of classes. Due to the simplicity of our assumptions, the bound remains quite raw and could be strengthened with further distributional assumptions on the means and variances.

Finally, to obtain a meaningful bound in Theorem C.2, we needed to assume $\Delta(\mathcal{K}) := \sup_{(\mu,\sigma^2)\in\mathcal{K}} \frac{q-\hat{q}}{q} < 1$. In Fig. C.2 we see that this can be assumed to hold on the compact sets on which logit means and variances tend to live (see also Fig. C.1).

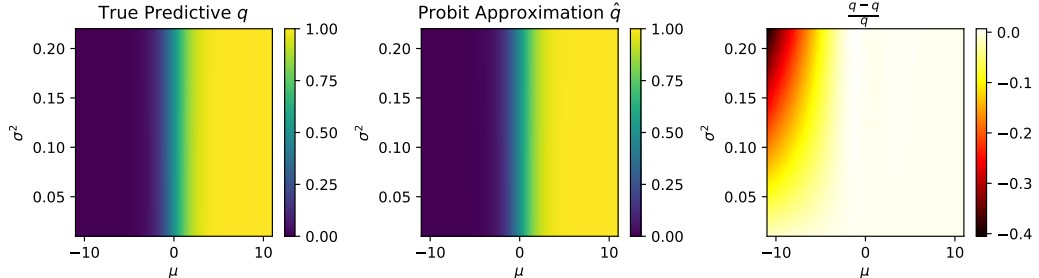

Figure C.2: Plot of the 'true' predictive $q$ (approximated with a $50000$ sample MC approximation) versus the probit approximation $\hat{q}$, as well as $\frac{q-\hat{q}}{q}$. We see that $\frac{q-\hat{q}}{q}$ tends to be negative, making the assumption $\sup_{(\mu,\sigma^2)\in\mathcal{K}} \frac{q-\hat{q}}{q} < 1$ from Theorem C.2 easily fulfilled.

Note that, in the proof of Theorem C.2, we did not use the form of the probit approximation explicitly, and the theorem holds for any approximation scheme $\hat{q}$ to the one dimensional integrals $q$.

# D    Moment Matching Beta Distributions

As noted in Section 4.2, when $\varphi = \Phi$ or $\rho$, we can construct a mapping

$$\mathrm{p}\colon \mathcal{G}(\mathbb{R}^C) \to \mathcal{B}((0,1))^C \tag{65}$$

by moment matching. Specifically, the parameters $\boldsymbol{\alpha}, \boldsymbol{\beta} \in (0,\infty)^C$ that match the moments of $\boldsymbol{Q} \sim \varphi_* \mathrm{f}$ for some $\mathrm{f} \in \mathcal{G}(\mathbb{R}^C)$ must satisfy

$$
\begin{aligned}
\mathbb{E}[\boldsymbol{Q}] &= \frac{\boldsymbol{\alpha}}{\boldsymbol{\alpha}+\boldsymbol{\beta}}, \\
\mathbb{E}[\boldsymbol{Q}^2] &= \frac{\boldsymbol{\alpha}(\boldsymbol{\alpha}+1)}{(\boldsymbol{\alpha}+\boldsymbol{\beta})(\boldsymbol{\alpha}+\boldsymbol{\beta}+1)},
\end{aligned}
\tag{66}
$$

where all vector operations are element-wise. Multiplying out the denominators on the right-hand side of the equations of Eq. (66), we obtain a system of two linear equations with two unknowns (for each $c$), which can be solved uniquely, yielding

$$
\begin{aligned}
\boldsymbol{\alpha} &:= \frac{\mathbb{E}[\boldsymbol{Q}] - \mathbb{E}[\boldsymbol{Q}^2]}{\mathbb{E}[\boldsymbol{Q}^2] - \mathbb{E}[\boldsymbol{Q}]^2}\mathbb{E}[\boldsymbol{Q}], \\
\boldsymbol{\beta} &:= \frac{\mathbb{E}[\boldsymbol{Q}] - \mathbb{E}[\boldsymbol{Q}^2]}{\mathbb{E}[\boldsymbol{Q}^2] - \mathbb{E}[\boldsymbol{Q}]^2}\left(1 - \mathbb{E}[\boldsymbol{Q}]\right).
\end{aligned}
\tag{67}
$$

which give us the parameters of the Beta distributions $\mathrm{p}(\mathrm{f})$.

## D.1 Information Geometric Interpretation of the Pushforward through NormCDF

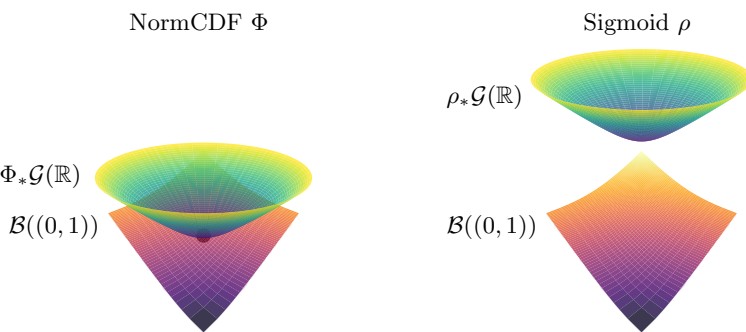

Figure D.1: Illustration of the statistical manifolds of the pushforward Gaussian distributions $\mathcal{G}(\mathbb{R})$ through the normCDF and the sigmoid respectively compared to the statistical manifold of Beta distributions $\mathcal{B}((0,1))$. NormCDF, unlike the sigmoid, makes the manifolds intersect at the point $\Phi_*\mathcal{N}(0,1) = B(1,1)$.

Here, we argue that the normCDF activation is an ideal choice for approximating Gaussian push-forwards with Beta distributions by interpreting Fig. D.1. This extends the work from [33], as it shows how one can make sense of the 'right' basis for performing Laplace approximations in the classification setting. Also see the discussion in [8].

$\Phi_*\mathcal{G}(\mathbb{R})$, $\rho_*\mathcal{G}(\mathbb{R})$ the space of pushforwards of Gaussian distributions by normCDF and sigmoid respectively, and $\mathcal{B}((0,1))$, the space of Beta distributions, are statistical manifolds naturally equipped with Riemannian metrics, that is their respective Fisher information metrics. We would like to visualise these manifolds. However, two difficulties arise.

The first difficulty is that, while these manifolds all lie in the infinite-dimensional vector space of signed measures on the open unit interval $\mathcal{M}((0,1))$, there is *no* subspace $\mathbb{V} \subset \mathcal{M}((0,1))$ which is 3-dimensional ($\mathbb{V} \cong \mathbb{R}^3$) and contains any two of these statistical manifolds ($\Phi_*\mathcal{G}(\mathbb{R}), \mathcal{B}((0,1)) \subset \mathbb{V}$ or $\rho_*\mathcal{G}(\mathbb{R}), \mathcal{B}((0,1)) \subset \mathbb{V}$). We will work around this by building distinct isometric embeddings $\Phi_*\mathcal{G}(\mathbb{R}) \hookrightarrow \mathbb{V}$, $\rho_*\mathcal{G}(\mathbb{R}) \hookrightarrow \mathbb{V}$ and $\mathcal{B}((0,1)) \hookrightarrow \mathbb{V}$ for some 3-dimensional vector space $\mathbb{V}$. This means that while the shape of the manifold illustrations is meaningful, the positioning of a manifold with respect to another is not, apart from some design choices that we describe below.

The second difficulty is that some–if not all–of these manifolds cannot be embedded isometrically into Euclidean space. As a workaround, we instead embed them into the 3-dimensional Minkowski space $\mathbb{R}^{2,1}$, that is $\mathbb{R}^3$ equipped with the pseudo-Riemannian metric $dx_1^2 + dx_2^2 - dx_3^2$.

The key observation is that $\Phi_*\mathcal{G}(\mathbb{R})$ and $\rho_*\mathcal{G}(\mathbb{R})$ are isometric to $\mathcal{G}(\mathbb{R})$. This is because $\Phi$ and $\rho$ are diffeomorphisms, so in particular sufficient statistics, and the Fisher information metric is invariant under sufficient statistics [3, Section 5.1.4]. Visually, this means that $\varphi_*\mathcal{G}(\mathbb{R})$ has the same shape irrespectively of the diffeomorphism activation function $\varphi$. One can thus observe that, given that $\mathcal{B}((0,1))$ is not isometric to $\mathcal{G}(\mathbb{R})$, there exists no activation $\varphi$ such that $\varphi_*\mathcal{G}(\mathbb{R}) = \mathcal{B}((0,1))$. To design an activation $\varphi$ that maps Gaussians to Betas, the best one can hope to do is to map one specific Gaussian distribution $\mathcal{N}(\mu,\sigma^2)$ to a specific Beta distribution $B(\alpha,\beta)$. This is done with the map $F_{\alpha,\beta}^{-1} \circ \Phi_{\mu,\sigma^2}$ where $\Phi_{\mu,\sigma^2}$ and $F_{\alpha,\beta}$ are the cumulative distribution functions of $\mathcal{N}(\mu,\sigma^2)$ and $B(\alpha,\beta)$ respectively. Taking $\mu = 0, \sigma^2 = 1, \alpha = \beta = 1$ we get $\Phi_{0,1} = \Phi$ and $F_{1,1} = \mathrm{id}_{(0,1)}$, yielding $F_{\alpha,\beta}^{-1} \circ \Phi_{\mu,\sigma^2} = \Phi$.

Now $\mathcal{G}(\mathbb{R})$, and hence $\Phi_*\mathcal{G}(\mathbb{R})$ and $\rho_*\mathcal{G}(\mathbb{R})$, is isometric to the hyperbolic plane Ay et al. [3, Example 3.1]. We can embed this isometrically into Minkowski space with the classical hyperboloid model of the hyperbolic plane [45],

$$\mathcal{G}(\mathbb{R}) \hookrightarrow \mathbb{R}^{2,1}. \tag{68}$$

For $\mathcal{B}((0,1))$, we use the isometric embedding from Le Brigant et al. [27, Proposition 2]:

$$\begin{aligned} \mathcal{B}((0,1)) &\hookrightarrow \mathbb{R}^{2,1} \\ (\alpha,\beta) &\mapsto (\eta(\alpha), \eta(\beta), \eta(\alpha+\beta)) \end{aligned} \tag{69}$$

where $\eta(a) := \int_1^a \sqrt{\psi'(r)} \, dr$ and $\psi$ is the digamma function.

Finally, we choose our embedding Eq. (68) for $\Phi_* \mathcal{G}(\mathbb{R})$ such that it intersects the embedding Eq. (69) at a point, to highlight that the statistical manifolds $\Phi_* \mathcal{G}(\mathbb{R})$ and $\mathcal{B}((0,1))$ intersect at a point in the infinite-dimensional ambient space $\mathcal{M}((0,1))$, while $\rho_* \mathcal{G}(\mathbb{R})$ and $\mathcal{B}((0,1))$ do not.

Moreover, since $\Phi_* \mathcal{N}(0,1) = B(1,1)$, our moment matching approximation Eq. (66) is exact when $f(x)$ is the standard Normal distribution.

## E  List of Predictive and Dirichlet Parameters Formulas

### E.1  Predictive Formulas

We gather formulas for all predictive estimators $\hat{p}$ of the true predictive $\mathbb{E}_{P \sim a_* \mathcal{N}(\mu, \Sigma)}[P]$, $\Sigma = \mathrm{diag}(\sigma^2)$, used in our experiments (Section 7).

#### E.1.1  Softmax

**Monte Carlo**  One can Monte Carlo estimate the true predictive as follows:

$$\hat{p} := \frac{1}{S} \sum_{s=1}^{S} \hat{p}^{(s)} \tag{70}$$

where the $\hat{p}^{(s)}$ are sampled i.i.d. from $\mathcal{N}(\mu, \Sigma)$.

**Mean Field (Appendix A.4)**  The Mean Field predictive [32] uses the following approximation for the true predictive:

$$\hat{p} := \mathrm{softmax}\left( \frac{\mu}{\sqrt{1 + \frac{\pi}{8} \sigma^2}} \right). \tag{71}$$

**Laplace Bridge**  The Laplace Bridge predictive [19] approximates the true predictive as follows:

$$\hat{p} := \frac{\frac{1}{\tilde{\sigma}^2}\left( 1 - \frac{2}{C} + \frac{e^{\tilde{\mu}}}{C^2} \sum_{c=1}^{C} e^{-\tilde{\mu}_c} \right)}{\sum_{c=1}^{C} \frac{1}{\tilde{\sigma}_c^2}\left( 1 - \frac{2}{C} + \frac{e^{\tilde{\mu}}}{C^2} \sum_{c'=1}^{C} e^{-\tilde{\mu}_{c'}} \right)} \tag{72}$$

where

$$\tilde{\mu}^2 := \sqrt{\frac{\sqrt{C/2}}{\sum_{c=1}^{C} \sigma_c^2}} \mu, \quad \tilde{\sigma}^2 := \frac{\sqrt{C/2}}{\sum_{c=1}^{C} \sigma_c^2} \sigma^2. \tag{73}$$

**Shekhovtsov & Flach.**  The Shekhovtsov and Flach [48] approximation of the Gaussian-softmax integral yields:

$$\hat{p} = \frac{n}{n + (A - b)^{s/s}} \tag{74}$$

where

$$A := \sum_{c=1}^{C} \exp(\tilde{\mu}_c / s), \quad n := \exp(\tilde{\mu}/s), \quad b := \exp(\tilde{\mu}/s), \tag{75}$$

$\tilde{\mu} := \mu - \max_c \mu_c$ for numerical stability,

$$\tau^2 := \sigma^2 + G, \quad \tau_{\min}^2 := \min_c \tau_c^2, \quad s := \sqrt{\frac{2\tau_{\min}^2}{L}}, \quad s := \sqrt{\frac{\tau^2 + \tau_{\min}^2}{L}} \tag{76}$$

and $L := \pi^2/3$, $G := L/2$.

### E.1.2 Exp

Our closed-form predictive for the exp activation function (Section 3.1) is given by

$$\hat{\boldsymbol{p}} := \frac{\exp\left(\boldsymbol{\mu} + \frac{\boldsymbol{\sigma}^2}{2}\right)}{\sum_{c=1}^{C}\exp\left(\mu_c + \frac{\sigma_c^2}{2}\right)}. \tag{77}$$

### E.1.3 NormCDF

The closed-form predictive for the normCDF activation function (Section 3.2) is computed as

$$\hat{\boldsymbol{p}} := \frac{\Phi\left(\frac{\boldsymbol{\mu}}{\sqrt{1+\boldsymbol{\sigma}^2}}\right)}{\sum_{c=1}^{C}\Phi\left(\frac{\mu_c}{\sqrt{1+\sigma_c^2}}\right)}. \tag{78}$$

### E.1.4 Sigmoid

For the sigmoid activation function (Section 3.2), the closed-form predictive can be computed as

$$\hat{\boldsymbol{p}} := \frac{\rho\left(\frac{\boldsymbol{\mu}}{\sqrt{1+\frac{\pi}{8}\boldsymbol{\sigma}^2}}\right)}{\sum_{c=1}^{C}\rho\left(\frac{\mu_c}{\sqrt{1+\frac{\pi}{8}\sigma_c^2}}\right)}. \tag{79}$$

## E.2 Dirichlet Parameters Formulas

We now gather the formulas of the parameters $\boldsymbol{\gamma}$ for the Dirichlet approximations to the Gaussian pushforwards.

### E.2.1 Softmax

The Laplace Bridge method [19] uses the following Dirichlet parameters:

$$\boldsymbol{\gamma} := \frac{1}{\tilde{\boldsymbol{\sigma}}^2}\left(1 - \frac{2}{C} + \frac{e^{\tilde{\boldsymbol{\mu}}}}{C^2}\sum_{c=1}^{C}e^{-\tilde{\mu}_c}\right) \tag{80}$$

where

$$\tilde{\boldsymbol{\mu}}^2 := \sqrt{\frac{\sqrt{C/2}}{\sum_{c=1}^{C}\sigma_c^2}}\boldsymbol{\mu}, \quad \tilde{\boldsymbol{\sigma}}^2 := \frac{\sqrt{C/2}}{\sum_{c=1}^{C}\sigma_c^2}\boldsymbol{\sigma}^2. \tag{81}$$

### E.2.2 Exp

Exp uses the closed-form parameters derived from Moment Matching the Gaussian pushforwards (Section 4.1):

$$\boldsymbol{\gamma} := \left(\prod_{c=1}^{C}\frac{a_c \cdot \max(\sum_{c'=1}^{C}a_{c'}, 1) - b_c}{b_c - a_c^2}\right)^{1/C}\frac{\boldsymbol{a}}{\sum_{c=1}^{C}a_c} \tag{82}$$

where

$$\boldsymbol{a} = \exp\left(\boldsymbol{\mu} + \frac{\boldsymbol{\sigma}^2}{2}\right), \quad \boldsymbol{b} = \exp(2\boldsymbol{\mu} + 2\boldsymbol{\sigma}^2). \tag{83}$$

### E.2.3 NormCDF

NormCDF uses the closed-form parameters derived from Moment Matching the Gaussian pushforwards (Section 4.1):

$$\boldsymbol{\gamma} := \left(\prod_{c=1}^{C}\frac{a_c \cdot \max(\sum_{c'=1}^{C}a_{c'}, 1) - b_c}{b_c - a_c^2}\right)^{1/C}\frac{\boldsymbol{a}}{\sum_{c=1}^{C}a_c} \tag{84}$$

where

$$a = \Phi\left(\frac{\boldsymbol{\mu}}{\sqrt{1+\boldsymbol{\sigma}^2}}\right), \ b = \Phi\left(\frac{\boldsymbol{\mu}}{\sqrt{1+\boldsymbol{\sigma}^2}}\right) - 2T\left(\frac{\boldsymbol{\mu}}{\sqrt{1+\boldsymbol{\sigma}^2}}, \frac{1}{\sqrt{1+2\boldsymbol{\sigma}^2}}\right). \tag{85}$$

### E.2.4 Sigmoid

Sigmoid uses the closed-form parameters derived from Moment Matching the Gaussian pushforwards (Section 4.1):

$$\boldsymbol{\gamma} := \left(\prod_{c=1}^{C}\frac{a_c \cdot \max(\sum_{c'=1}^{C}a_{c'},1) - b_c}{b_c - a_c^2}\right)^{1/C}\frac{\boldsymbol{a}}{\sum_{c=1}^{C}a_c} \tag{86}$$

where

$$
\begin{aligned}
a &= \rho\left(\frac{\boldsymbol{\mu}}{\sqrt{1+\frac{\pi}{8}\boldsymbol{\sigma}^2}}\right), \\
b &= \rho\left(\frac{\boldsymbol{\mu}}{\sqrt{1+\frac{\pi}{8}\boldsymbol{\sigma}^2}}\right) - \frac{1}{\sqrt{1+\frac{\pi}{8}\boldsymbol{\sigma}^2}}\rho\left(\frac{\boldsymbol{\mu}}{\sqrt{1+\frac{\pi}{8}\boldsymbol{\sigma}^2}}\right)\left(1 - \rho\left(\frac{\boldsymbol{\mu}}{\sqrt{1+\frac{\pi}{8}\boldsymbol{\sigma}^2}}\right)\right).
\end{aligned}
\tag{87}
$$

## F List of Uncertainty Estimators

In this section, we list the uncertainty estimators used in our experiments (Section 7).

### F.1 Predictive

Given a predictive $p$, we consider two uncertainty estimators.

**Maximum Probability**

$$\arg\max_{c\in\{1,\dots C\}} p_c. \tag{88}$$

**Entropy**

$$-\sum_{c=1}^{C}p_c \log p_c. \tag{89}$$

### F.2 Monte Carlo

Given $S$ Monte Carlo samples $\hat{\boldsymbol{p}}^{(1)},\dots,\hat{\boldsymbol{p}}^{(S)}$ with mean $\hat{\boldsymbol{p}}$, one can calculate a predictive as their average and derive the estimators in Appendix F.1. However, Monte Carlo samples allow one to calculate two additional estimators detailed below.

**Expected Entropy**

$$-\frac{1}{S}\sum_{s=1}^{S}\sum_{c=1}^{C}\hat{p}_c^{(s)}\log\hat{p}_c^{(s)}. \tag{90}$$

**Mutual Information/Jensen-Shannon Divergence**

$$-\sum_{c=1}^{C}\hat{p}_c\log\hat{p}_c + \frac{1}{S}\sum_{s=1}^{S}\sum_{c=1}^{C}\hat{p}_c^{(s)}\log\hat{p}_c^{(s)}. \tag{91}$$

### F.3 Dirichlet

Given a second-order Dirichlet distribution with parameters $\boldsymbol{\gamma}$, one can obtain the expected entropy and mutual information estimators without the need for Monte Carlo samples.

**Expected Entropy**

$$-\sum_{c=1}^{C}\frac{\gamma_c}{\sum_{c'=1}^{C}\gamma_{c'}}\left(\psi(\gamma_c+1) - \psi\left(\sum_{c'=1}^{C}\gamma_{c'}+1\right)\right) \tag{92}$$

where $\psi$ is the digamma function.

**Mutual Information**

$$\sum_{c=1}^{C} \frac{\gamma_c}{\sum_{c'=1}^{C} \gamma_{c'}} \left( -\log \gamma_c + \log \left( \sum_{c'=1}^{C} \gamma_{c'} \right) + \psi(\gamma_c + 1) - \psi \left( \sum_{c'=1}^{C} \gamma_{c'} + 1 \right) \right). \quad (93)$$

# G   Experimental Setup

This section describes our experimental setup in detail.

We have two main research questions:

- What are the effects of changing the learning objective?
- Do we have to sacrifice performance for sample-free predictives?

To answer the first question, we evaluate our closed-form predictives (Sigmoid, NormCDF, Exp) and moment-matched Dirichlet distributions against softmax models equipped with approximate inference tools (Laplace Bridge [19], Mean Field [32], Monte Carlo sampling). We consider **Heteroscedastic Classifiers (HET)** [7], **Spectral-Normalised Gaussian Processes (SNGP)** [31], and last-layer **Laplace approximation** methods [11] as backbones (see Appendix J for details).

The resulting 21 (method, activation, predictive) triplets are evaluated on ImageNet-1k [12] and CIFAR-10 [26] on five metrics aligning with practical needs from uncertainty estimates [39]:

1. Log probability proper scoring rule for the predictive,
2. Expected calibration error of the predictive's maximum-probability confidence,
3. Binary log probability proper scoring rule for the correctness prediction task,
4. Accuracy of the predictive's argmax,
5. AUROC for the out-of-distribution (OOD) detection task.

See Appendix I for details.

For ImageNet, we treat ImageNet-C [18] samples with 15 corruption types and 5 severity levels as OOD samples. For CIFAR-10, we use the CIFAR-10C corruptions.

For the second question, we consider fixed (method, activation) pairs and test whether our methods perform on par with the Monte Carlo sampled predictives.

To provide a fair comparison, we reimplement each method as simple-to-use wrappers around deterministic backbones.

For ImageNet evaluation, we use a ResNet-50 backbone pretrained with the softmax activation function, and train each (method, activation) pair for 50 ImageNet-1k epochs following Mucsányi et al. [40]. We train with the LAMB optimiser [57] using a batch size of 128 and gradient accumulation across 16 batches, resulting in an effective batch size of 2048, following Tran et al. [52]. We further use a cosine learning rate schedule with a single warmup epoch using a warmup learning rate of $0.0001$. The learning rate is treated as a hyperparameter and selected from the interval $[0.0005, 0.05]$ based on the validation performance. The weight decay is selected from the set $\{0.01, 0.02\}$. During training, we keep track of the best-performing checkpoint on the validation set and load it before testing. We search for ideal hyperparameters with a ten-step Bayesian Optimization scheme [47] in Weights & Biases [5] based on the negative log-likelihood.

On CIFAR-10, we train ResNet-28 models from scratch for 100 epochs. The only exceptions are the SNGP models that are trained for 125 epochs [31]. We train with Momentum SGD using a batch size of 128 and no gradient accumulation. Similarly to ImageNet, we use a cosine learning rate schedule but with five warmup epochs and warmup learning rate $1e{-}5$. The learning rate is also treated as a hyperparameter on CIFAR-10. We use the interval $[0.05, 1]$ for Sigmoid and NormCDF, and $[0.01, 0.15]$ for Softmax and Exp. The optimal learning rates for Sigmoid and NormCDF are generally larger, as the class-wise binary cross-entropies are averaged instead of summed. The weight decay is selected from the interval $[1e{-}6, 1e{-}4]$. Similarly to ImageNet, we

Table H.1: Runtime overhead of Monte Carlo sampling on ImageNet as a percentage of total inference time across models and batch sizes.

| Batch Size | Mean Field | 1 sample | 10 samples | 100 samples | 1000 samples |
|---|---|---|---|---|---|
| **ResNet-26** (C=10) | | | | | |
| 1 | 2.75% | 9.82% | 10.63% | 10.70% | 10.70% |
| 16 | 2.04% | 10.61% | 11.51% | 11.51% | 11.92% |
| 32 | 2.11% | 10.98% | 11.94% | 11.96% | 12.42% |
| 64 | 2.11% | 10.88% | 11.90% | 11.95% | 12.34% |
| 128 | 2.04% | 8.12% | 8.84% | 8.96% | 9.61% |
| 256 | 1.17% | 4.31% | 4.73% | 4.93% | 7.14% |
| **Wide-ResNet-26-5** (C=100) | | | | | |
| 1 | 1.29% | 8.54% | 9.36% | 9.40% | 9.28% |
| 16 | 1.16% | 5.94% | 6.58% | 6.84% | 8.56% |
| 32 | 0.86% | 3.36% | 3.76% | 3.98% | 6.95% |
| 64 | 0.47% | 1.83% | 2.03% | 2.12% | 5.82% |
| 128 | 0.23% | 0.90% | 1.04% | 1.04% | 4.88% |
| 256 | 0.12% | 0.46% | 0.53% | 0.84% | 4.47% |
| **ResNet-50** (C=1000) | | | | | |
| 1 | 1.15% | 5.43% | 5.88% | 6.09% | 6.30% |
| 16 | 0.32% | 1.25% | 1.43% | 1.70% | 8.04% |
| 32 | 0.16% | 0.64% | 0.72% | 1.34% | 7.51% |
| 64 | 0.09% | 0.36% | 0.41% | 1.13% | 7.75% |
| 128 | 0.05% | 0.20% | 0.22% | 1.00% | 7.82% |
| 256 | 0.03% | 0.10% | 0.18% | 0.95% | 8.05% |

use the best-performing checkpoint in the tests and use a ten-step Bayesian Optimization scheme to select performant hyperparameters.

The hyperparameter optimization, training, and evaluation of the methods used in this paper took 0.8 GPU years on NVIDIA RTX 2080Ti GPUs in a university compute cluster. The individual runs required no more than 50 GB of RAM and 3 days of runtime.

## H    Runtime Overhead of Monte Carlo Sampling

We measure the time to obtain the predictives from the logit-space Gaussians compared to the time of a forward pass on an NVIDIA RTX 2080 Ti GPU (with similar results on Tesla A100 GPUs). As shown in Table H.1, ResNet-50 with 1000 samples shows 6-8% overhead depending on the batch size, confirming our claim in the main text. In contrast, the runtime cost of computing our closed-form predictives is negligible compared to a forward pass, especially for larger batch sizes. These results demonstrate that even with a diagonal covariance structure, MC sampling yields a nontrivial overhead, especially for large numbers of classes and samples. In comparison, our closed-form approach has negligible computational cost.

## I    Benchmark Metrics

Our experiments use five tasks/metrics:

1. Log probability proper scoring rule for the predictive,
2. Expected calibration error of the predictive's maximum-probability confidence,
3. Binary log probability proper scoring rule for the correctness prediction task,
4. Accuracy of the predictive's argmax,
5. AUROC for the out-of-distribution detection task.

Below, we describe these metrics and their respective tasks.

## I.1  Log Probability Proper Scoring Rule for the Predictive

First, we briefly discuss proper and strictly proper scoring rules over general probability measures based on [39].

Consider a function $S\colon \mathcal{Q} \times \mathcal{Y} \to \mathbb{R}$ where $\mathcal{Q}$ is a family of probability distributions over the space $\mathcal{Y}$, called the label space.

$S$ is called a proper scoring rule if and only if

$$\max_{q \in \mathcal{Q}} \mathbb{E}_{Y \sim p} S(q, Y) = \mathbb{E}_{Y \sim p} S(p, Y), \tag{94}$$

i.e., $p$ is *one of* the maximisers of $S$ in $q$ in expectation. $S$ is further *strictly* proper if $\arg\max_{q \in \mathcal{Q}} \mathbb{E}_{Y \sim p} S(q, Y) = p$ is the *unique* maximiser of $S$ in $q$ in expectation.

The log probability scoring rule for categorical distributions is defined as

$$S(q, c) = \sum_{c'=1}^{C} \delta_{c,c'} \log q_{c'}(x) = \log q_c(x), \tag{95}$$

where $c \in \{1, \ldots, C\}$ is the true class and $\delta$ is the Kronecker delta. $S$ defined this way is a strictly proper scoring rule, i.e. $\mathbb{E}_{Y \sim p} S(q, Y)$ is maximal if and only if

$$q(Y = c \mid x) = p(Y = c \mid x) \ \forall c \in \{1, \ldots, C\}. \tag{96}$$

The score above is equivalent to the negative cross-entropy loss.

## I.2  Expected Calibration Error

To set up the required quantities for the Expected Calibration Error (ECE) metric [42], we follow the steps below, based on [39].

1. Train a neural network on the training dataset.

2. Create predictions and confidence estimates on the test data.

3. Group the predictions into $M$ bins based on the confidences estimates. Define bin $B_m$ to be the set of all indices $n$ of predictions $(\hat{y}_n, \tilde{c}_n)$ for which

$$\tilde{c}_n \in \left( \frac{m-1}{M}, \frac{m}{M} \right]. \tag{97}$$

The Expected Calibration Error (ECE) metric [42] is then defined as

$$\mathrm{ECE} = \sum_{m=1}^{M} \frac{|B_m|}{n} \left| \mathrm{acc}(B_m) - \mathrm{conf}(B_m) \right| \tag{98}$$

where

$$\mathrm{acc}(B_m) = \frac{1}{|B_m|} \sum_{n \in B_m} \mathbb{1}\left( \hat{y}_n = c_n \right), \tag{99}$$

$$\mathrm{conf}(B_m) = \frac{1}{|B_m|} \sum_{n \in B_m} \max_{1 \le c \le C} f_c(x_n). \tag{100}$$

Intuitively, the ECE is high when the model's per-bin confidences match its accuracy on the bin. We use $M = 15$ bins in this paper.

## I.3  Binary Log Probability Proper Scoring Rule for Correctness Prediction

The correctness prediction task measures the models' ability to predict the correctness of their own predictions. We consider *correctness estimators* $\tilde{c}(x) \in [0, 1]$ for inputs $x \in \mathcal{X}$ derived from the predictives. Framed as a binary prediction task, the goal of these estimators is to predict the

probability of the predicted class' correctness. In particular, for an (input, target) pair $(x, y)$ with $x \in \mathcal{X}, y \in \mathcal{Y}$, we set the correctness target to

$$\ell \equiv \ell(x, y) = 1 \left( \max_{1 \leq c \leq C} h_c(x) = y \right). \tag{101}$$

Dropping the dependency on $x \in \mathcal{X}$ for brevity, the log probability score for binary targets $\ell \in \{0, 1\}$ and estimators $\tilde{c} \in [0, 1]$ is defined as

$$S(\tilde{c}, \ell) = \begin{cases} \log c & \text{if } \ell = 1 \\ \log(1 - \tilde{c}) & \text{if } \ell = 0 \end{cases} = \ell \log \tilde{c} + (1 - \ell) \log(1 - \tilde{c}). \tag{102}$$

One can show that this is indeed a strictly proper scoring rule [39].

## I.4 Accuracy

For completeness, the accuracy of a predictive $h$ on a dataset $(x_n, c_n)_{n=1}^N$ is

$$\mathrm{acc}(h; (x_n, c_n)_{n=1}^N) = \frac{1}{N} \sum_{n=1}^N 1 \left( \arg\max_{c \in \{1, \dots, C\}} h_c(x_n) = \tilde{c}_n \right). \tag{103}$$

## I.5 Area Under the Receiver Operating Characteristic Curve for Out-of-Distribution Detection

Out-of-distribution detection is another binary prediction task where a general uncertainty estimator $u(x) \in \mathbb{R}$ (derived from predictives or second-order Dirichlet distributions) is tasked to separate ID and OOD samples from a balanced mixture. The target OOD indicator variable $o(x)$ is, therefore, binary. As the uncertainty estimator can take on any real value, we measure the Area Under the Receiver Operating Characteristic curve (AUROC), which quantifies the separability of ID and OOD samples w.r.t. the uncertainty estimator.

Given uncertainty estimates $u_n \equiv u(x_n)$ and target binary labels $o_i$ on a balanced dataset $(x_n, o_n)_{n=1}^N$, as well as a threshold $t \in \mathbb{R}$, we predict 1 (out-of-distribution) when $u_n \geq t$ and 0 (in-distribution) when $u_n < t$. This lets us define the following index sets:

$$\begin{aligned} \text{True positives: } \mathrm{TP}(t) &= \{n : o_n = 1 \wedge u_n \geq t\} \\ \text{False positives: } \mathrm{FP}(t) &= \{n : o_n = 0 \wedge u_n \geq t\} \\ \text{False negatives: } \mathrm{FN}(t) &= \{n : o_n = 1 \wedge u_n < t\} \\ \text{True negatives: } \mathrm{TN}(t) &= \{n : o_n = 0 \wedge u_n < t\}. \end{aligned} \tag{104}$$

The Receiver Operating Characteristic (ROC) curve compares the following quantities:

$$\begin{aligned} \mathrm{TPR}(t) &= \frac{|\mathrm{TP}(t)|}{|\mathrm{TP}(t)| + |\mathrm{FN}(t)|} = \frac{|\mathrm{TP}(t)|}{|\mathrm{P}|} \\ \mathrm{FPR}(t) &= \frac{|\mathrm{FP}(t)|}{|\mathrm{FP}(t)| + |\mathrm{TN}(t)|} = \frac{|\mathrm{FP}(t)|}{|\mathrm{N}|}. \end{aligned} \tag{105}$$

Here, FPR tells us how many of the actual negative samples in the dataset are recalled (predicted positive) at threshold $t$.

One can draw a curve of $(\mathrm{FPR}(t), \mathrm{TPR}(t))$ for all $t$ from $-\infty$ to $+\infty$. This is the *ROC curve*. The area under this curve quantifies how well the uncertainty estimator $u(x)$ can separate in-distribution and out-of-distribution inputs.

# J  Benchmarked Methods

This section describes our benchmarked methods and provides further implementation details.

## J.1 Spectral Normalised Gaussian Process

Spectral normalised Gaussian processes (SNGP) [31] use spectral normalization of the parameter tensors for distance-awareness and a last-layer Gaussian process approximated by Fourier features to capture uncertainty. For an input $x \in \mathcal{X}$ and number of classes $C$, they predict a $C$-variate Gaussian distribution

$$\mathcal{N}\left(\boldsymbol{B}\boldsymbol{\phi}(x), \boldsymbol{\phi}(x)^\top \left(\boldsymbol{\Psi}^\top \boldsymbol{\Psi} + \boldsymbol{I}\right)^{-1} \boldsymbol{\phi}(x)\boldsymbol{I}_C\right) \tag{106}$$

in logit space.

- $\boldsymbol{B} \in \mathbb{R}^{C \times D}$ is a *learned* parameter matrix that maps pre-logits to logits.
- $\boldsymbol{\phi}(x) = \cos\left(\boldsymbol{W}\boldsymbol{f}^{L-1}(x) + \boldsymbol{b}\right) \in \mathbb{R}^D$ is a random pre-logit embedding of the input $x \in \mathcal{X}$. $\boldsymbol{f}^{L-1}(x)$ denotes the pre-logit embedding. $\boldsymbol{W}$ is a *fixed* semi-orthogonal random matrix, and $\boldsymbol{b}$ is also a *fixed* random vector but sampled from Uniform$(0, 2\pi)$.
- $\boldsymbol{\Psi}^\top \boldsymbol{\Psi}$ is the (unnormalised) empirical covariance matrix of the pre-logits of the training set. This is calculated by accumulating the mini-batch estimates during the last epoch.[4]

The method applies spectral normalization to the hidden weights in each layer using a power iteration scheme with a single iteration per batch to obtain the largest singular value. Liu et al. [31] claim this helps with input distance awareness.

## J.2 Heteroscedastic Classifier

Heteroscedastic classifiers (HET) [7] construct a Gaussian distribution in the logit space to model per-input uncertainties:

$$\mathcal{N}(\boldsymbol{f}(x), \boldsymbol{\Sigma}(x)), \tag{107}$$

where $\boldsymbol{f}(x) \in \mathbb{R}^D$ is the logit mean for input $x \in \mathcal{X}$ and

$$\boldsymbol{\Sigma}(x) = \boldsymbol{V}(x)^\top \boldsymbol{V}(x) + \mathbf{diag}(\boldsymbol{d}(x)) \tag{108}$$

is a (positive definite) covariance matrix. Both the low-rank term $\boldsymbol{V}(x)$ and the diagonal term $\boldsymbol{d}(x)$ are calculated as a linear function of the pre-logit layer's output.

To learn the per-input covariance matrices from the training set, one has to construct a predictive estimate from $\mathcal{N}(\boldsymbol{f}(x), \boldsymbol{\Sigma}(x))$ using any of the methods in Appendix E. This predictive estimate is then trained using a standard cross-entropy (NLL) loss.

HET uses a temperature parameter to scale the logits before calculating the BMA. This is chosen using a validation set.

The off-diagonal terms of the covariance matrix do not affect the approximate predictive (Eq. (12)). This means that, in our framework, one can discard the low-rank term $\boldsymbol{V}(x)$ and only model the diagonal term $\boldsymbol{d}(x)$ without a decrease in expressivity. To keep comparisons fair and use the same backbone with the same number of parameters, we also only model $\boldsymbol{d}(x)$ for softmax-based predictives.

## J.3 Laplace Approximation

The Laplace approximation [11] approximates the posterior $p(\boldsymbol{\theta} \mid \mathcal{D})$ over the network parameters $\boldsymbol{\theta}$ for a Gaussian prior $p(\boldsymbol{\theta})$ and likelihood defined by the network architecture by a Gaussian. In its simplest form, it uses the maximum a posteriori (MAP) weights $\boldsymbol{\theta}_{\text{MAP}} \in \mathbb{R}^P$ as the mean and the inverse Hessian of the *regularised* loss over the training set $\tilde{\mathcal{L}}(\boldsymbol{\theta}; \mathcal{D}) = \mathcal{L}(\boldsymbol{\theta}; \mathcal{D}) + \lambda \|\boldsymbol{\theta}\|_2^2$ evaluated at the MAP as the covariance matrix:

$$\mathcal{N}\left(\boldsymbol{\theta}_{\text{MAP}}, \left(\left.\frac{\partial^2 \tilde{\mathcal{L}}(\boldsymbol{\theta}; \mathcal{D})}{\partial \theta_i \partial \theta_j}\right|_{\boldsymbol{\theta}_{\text{MAP}}}\right)^{-1}\right) = \mathcal{N}\left(\boldsymbol{\theta}_{\text{MAP}}, \left(\left.\frac{\partial^2 \mathcal{L}(\boldsymbol{\theta}; \mathcal{D})}{\partial \theta_i \partial \theta_j}\right|_{\boldsymbol{\theta}_{\text{MAP}}} + \lambda \boldsymbol{I}_P\right)^{-1}\right). \tag{109}$$

---

[4]As we use a cosine learning rate decay in all experiments, the model makes negligible changes in its pre-logit feature space in the last epoch. Thus, the empirical covariance matrix is approximately consistent.

This is a *locally optimal* post-hoc Gaussian approximation of the true posterior $p(\boldsymbol{\theta} \mid \mathcal{D})$ based on a second-order Taylor approximation. For details, see [50].

For deep neural networks, the Hessian matrix is often replaced with the Generalised Gauss-Newton (GGN) matrix. The GGN is guaranteed to be positive semidefinite even for suboptimal weights and has efficient approximation schemes, such as Kronecker-Factored Approximate Curvature [37] or low-rank approximations.

Denoting our curvature estimate of choice by $\boldsymbol{G}$, the logit-space Gaussian is obtained by pushing forward the weight-space Gaussian measure through the *linearised* model around $\boldsymbol{\theta}_{\text{MAP}}$. For an input $x \in \mathcal{X}$, this results in

$$\mathcal{N}\left(\boldsymbol{f}(x, \boldsymbol{\theta}_{\text{MAP}}), (\boldsymbol{J}_{\boldsymbol{\theta}_{\text{MAP}}}\boldsymbol{f}(x))\left(\boldsymbol{G} + \lambda \boldsymbol{I}_P\right)^{-1}\left(\boldsymbol{J}_{\boldsymbol{\theta}_{\text{MAP}}}\boldsymbol{f}(x)\right)^{\top}\right), \tag{110}$$

where $\boldsymbol{J}_{\boldsymbol{\theta}_{\text{MAP}}}\boldsymbol{f}(x) \in \mathbb{R}^{C \times P}$ is the model Jacobian matrix.

We use a last-layer KFAC Laplace variant in our experiments and use the *full* training set for calculating the GGN instead of a mini-batch based on recent works on the bias in mini-batch estimates [50].

## K  Additional Results

### K.1  Closed-Form Softmax Predictives

One can directly apply Eq. (10) to a neural network trained with the softmax cross-entropy loss in Eq. (4) to obtain closed-form predictives. However, we empirically found this approach to decrease performance, as the denominator of Eq. (10) increased to the order of billions during training. Considering a multivariate normal random variable $\boldsymbol{x} \sim \mathcal{N}(\boldsymbol{\mu}, \Sigma)$ representing a logit distribution,

$$\text{Var}\left(\sum_{i=1}^{C} e^{x_i}\right) \geq \text{Var}(e^{x_j}) = e^{2\mu_j + \Sigma_{jj}}\underbrace{(e^{\Sigma_{jj}} - 1)}_{>0}, \tag{111}$$

i.e., when $e^{\mu_j}$ is large for some $j \in \{1, \ldots, C\}$, the variance of the denominator necessarily explodes, violating our assumption.

To mitigate this issue, one may optimise the regularised cross-entropy loss

$$\mathcal{L}((x_n, c_n)_{n=1}^{N}) = \tilde{\mathcal{L}}((x_n, c_n)_{n=1}^{N}) + \lambda \sum_{n=1}^{N}\left(\sum_{c=1}^{C}\exp(f_c(x_n)) - 1\right)^2 \tag{112}$$

or the more numerically stable version

$$\mathcal{L}((x_n, c_n)_{n=1}^{N}) = \tilde{\mathcal{L}}((x_n, c_n)_{n=1}^{N}) + \lambda \sum_{n=1}^{N}\left(\log\sum_{c=1}^{C}\exp(f_c(x_n))\right)^2 \tag{113}$$

for an appropriately tuned $\lambda$ hyperparameter. The latter formulation can be stably trained even at an ImageNet scale. However, there are fundamental limitations to the predictive given by Eq. (10). Namely, for *isotropic* logit-space Gaussian distributions, $\sigma^2(x)/2$ introduces a constant shift in the logits, under which the softmax activation function is invariant. Therefore, the predictive approximation collapses into the MAP prediction. The SNGP method predicts such isotropic logit-space Gaussians. Further, we empirically found that the Laplace method's predictives for models trained with the regularised cross-entropy loss were also approximately isotropic; thus, this integral approximation yielded diminishing returns.

For completeness, we share the ImageNet and CIFAR-10 results of $\varphi = \exp$ in Tables K.1 and K.2.

### K.2  Vision Transformer Results on ImageNet

Table K.3 shows results on ViT Little [14] backbones from the `timm` [53] library on the ImageNet validation set. As in the main paper, the hyperparameters are optimised using a ten-step Bayesian hyperparameter optimization scheme of Weights and Biases.

Table K.1: Performance metrics of $\varphi = \exp$ on the CIFAR-10 validation dataset. We report the mean and two standard deviations.

| Metric | Exp SNGP | Exp HET | Exp Laplace |
|---|---|---|---|
| NLL | $0.376 \pm 0.007$ | $0.353 \pm 0.059$ | $0.370 \pm 0.024$ |
| ECE | $0.041 \pm 0.006$ | $0.029 \pm 0.007$ | $0.045 \pm 0.007$ |
| Log Prob. | $-0.255 \pm 0.005$ | $-0.238 \pm 0.038$ | $-0.255 \pm 0.016$ |
| Accuracy | $0.884 \pm 0.013$ | $0.888 \pm 0.023$ | $0.893 \pm 0.007$ |
| AUROC | $0.592 \pm 0.006$ | $0.603 \pm 0.009$ | $0.596 \pm 0.011$ |

Table K.2: Performance metrics of $\varphi = \exp$ on the ImageNet validation dataset. We report the mean and two standard deviations.

| Metric | Exp SNGP | Exp HET | Exp Laplace |
|---|---|---|---|
| NLL | $0.866 \pm 0.031$ | $0.929 \pm 0.042$ | $0.986 \pm 0.149$ |
| ECE | $0.058 \pm 0.004$ | $0.022 \pm 0.007$ | $0.058 \pm 0.013$ |
| Log Prob. | $-0.380 \pm 0.009$ | $-0.403 \pm 0.011$ | $-0.402 \pm 0.014$ |
| Accuracy | $0.784 \pm 0.001$ | $0.779 \pm 0.001$ | $0.785 \pm 0.002$ |
| AUROC | $0.606 \pm 0.001$ | $0.611 \pm 0.001$ | $0.615 \pm 0.003$ |

## K.3 CIFAR-10 Results

This appendix section repeats the experiments presented in the main paper on the CIFAR-10 dataset. For a detailed description of the experimental setup, refer to Appendix G. Appendix I describes the used tasks and metrics.

As stated in the main paper, our two research questions are:

- What are the effects of changing the learning objective? (Appendix K.3.2)
- Do we have to sacrifice performance for sample-free predictives? (Appendix K.3.1)

### K.3.1 Quality of Sample-Free Predictives

Similarly to the main paper, in this section, we investigate our first research question: whether there is a price to pay for sample-free predictives. Table K.4 showcases the two best-performing (activation, method) pairs on the ECE metric and the CIFAR-10 dataset: Softmax and NormCDF Laplace. Mean Field (MF) is a strong alternative for sample-free predictives, but it has no guarantees and can fall

Table K.3: NLL and accuracy metrics of ViT Little models on the ImageNet validation set. Error bars represent two standard deviations.

| Method | NLL | Accuracy (%) |
|---|---|---|
| **Laplace** | | |
| Softmax | $0.81988 \pm 0.0020$ | $79.598 \pm 0.145$ |
| NormCDF | $0.79603 \pm 0.0087$ | $80.678 \pm 0.042$ |
| Sigmoid | $0.79479 \pm 0.0002$ | $80.104 \pm 0.206$ |
| **HET** | | |
| Softmax | $0.87096 \pm 0.0242$ | $78.604 \pm 0.378$ |
| NormCDF | $0.79106 \pm 0.0075$ | $80.514 \pm 0.425$ |
| Sigmoid | $0.88175 \pm 0.0356$ | $78.738 \pm 0.324$ |
| **GP** | | |
| Softmax | $0.86724 \pm 0.0229$ | $78.898 \pm 0.187$ |
| NormCDF | $0.76974 \pm 0.0088$ | $80.694 \pm 0.575$ |
| Sigmoid | $0.81178 \pm 0.0115$ | $79.980 \pm 0.412$ |

Table K.4: Comparison of ECE results for different predictives using a fixed Laplace backbone.

| Method | Mean $\pm$ 2 std |
|---|---|
| **Softmax Laplace** | |
| MC 100 | $0.0096 \pm 0.0013$ |
| MC 1000 | $0.0102 \pm 0.0016$ |
| MC 10 | $0.0120 \pm 0.0013$ |
| Mean Field | $0.0121 \pm 0.0029$ |
| Laplace Bridge Predictive | $0.5933 \pm 0.0105$ |
| **NormCDF Laplace** | |
| Closed-form | $0.0074 \pm 0.0012$ |
| MC 1000 | $0.0092 \pm 0.0022$ |
| MC 100 | $0.0095 \pm 0.0015$ |
| MC 10 | $0.0100 \pm 0.0020$ |

Figure K.1: CIFAR-10 ECE results. Our closed-form predictives (■) outperform Softmax (■) on HET and SNGP. Laplace tunes its hyperparameters based on the ECE metric – NormCDF Laplace is the overall best method. Note the restricted $y$-limits for readability.

behind MC sampling (see also Fig. 1). Empirically, our closed-form predictives always perform on par with MC sampling.

### K.3.2 Effects of Changing the Learning Objective

As in the main paper, in this section, we use the best-performing predictive and estimator (see Appendix E) for softmax models and employ our methods with the closed-form predictives.

**Calibration and Proper Scoring** We first evaluate calibration using the log probability scoring rule [17] and the Expected Calibration Error (ECE) metric [42]. Fig. 3b shows that on CIFAR-10, the score of our closed-form predictives (Sigmoid, NormCDF) are consistently better than the corresponding softmax results for all methods.

Fig. K.1 shows that on CIFAR-10, our closed-form predictives have a clear advantage on HET and SNGP. Laplace is a post-hoc method that tunes its hyperparameters on the ECE metric, hence its enhanced performance. Our NormCDF predictive is on par with Softmax.

**Correctness Prediction** Fig. K.2 shows that on the correctness prediction task, our closed-form predictives outperform all Softmax predictives across all methods, as measured by the log probability proper scoring rule.

**Accuracy** Closed-form predictives do not sacrifice accuracy. Fig. K.3 evidences this claim on CIFAR-10: our closed-form predictives either outperform or are on par with Softmax predictives. The most accurate method is Sigmoid HET. These results support the findings of Wightman et al. [54] that showcase desirable training dynamics of the class-wise cross-entropy loss.

**Out-of-Distribution Detection** Finally, we consider the OOD detection task on a balanced mixture of ID (CIFAR-10) and OOD inputs. As OOD inputs, we consider corrupted CIFAR-10C samples. We use the AUROC metric to evaluate the methods' performance. As shown in Fig. K.4, the best-performing method is Sigmoid SNGP, a closed-form method. Generally, Softmax performs on par

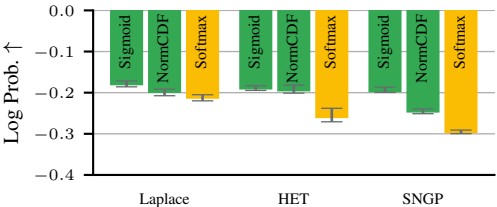

Figure K.2: CIFAR-10 log probability proper scoring results for the binary correctness prediction task. Our closed-form predictives (■) consistently outperform Softmax (■) on all methods.

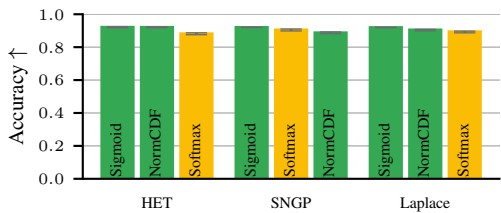

Figure K.3: **Closed-form predictives do not sacrifice accuracy.** CIFAR-10 accuracies. Our closed-form predictives (■) either outperform or are on par with Softmax (■) across all methods.

with our closed-form predictives. Intuitively, separating ID and OOD samples does not require a fine-grained representation of uncertainty, unlike the ECE or proper scoring rules. Nevertheless, the closed-form predictives and second-order Dirichlet distributions are considerably cheaper to calculate than Softmax MC predictions (see Section 6).

## K.4 CIFAR-100 Results

We repeat the NLL and accuracy evaluation on CIFAR-100 using WideResNet-28-5 models following a ten-step Bayesian hyperparameter sweep on Weights and Biases. Results are shown in Table K.5.

Table K.5: NLL and accuracy metrics of WideResNet-28-5 models on the CIFAR-100 test set. Error bars represent two standard deviations.

| Method | NLL | Accuracy (%) |
|---|---|---|
| **Laplace** | | |
| Softmax | $0.92176 \pm 0.0128$ | $78.225 \pm 0.412$ |
| NormCDF | $0.88823 \pm 0.0094$ | $78.900 \pm 0.325$ |
| Sigmoid | $0.94209 \pm 0.0136$ | $78.050 \pm 0.436$ |
| **HET** | | |
| Softmax HET | $0.98513 \pm 0.0254$ | $77.050 \pm 0.228$ |
| NormCDF HET | $0.94643 \pm 0.0022$ | $78.117 \pm 0.286$ |
| Sigmoid HET | $0.95251 \pm 0.0142$ | $77.583 \pm 0.175$ |
| **SNGP** | | |
| Softmax SNGP | $0.97156 \pm 0.0348$ | $77.750 \pm 0.462$ |
| NormCDF SNGP | $0.90011 \pm 0.0106$ | $79.010 \pm 0.298$ |
| Sigmoid SNGP | $0.99996 \pm 0.0265$ | $78.350 \pm 0.215$ |

## K.5 Alignment with the True Predictives

Table K.6 shows the divergence of predictives using various approximation techniques from the true predictive estimated using $10{,}000$ Monte Carlo samples. Our closed-form predictives, NormCDF and Sigmoid, are favorable to both the mean field and Laplace bridge approximations.

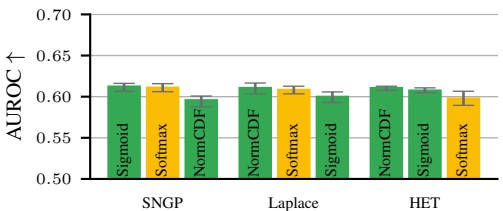

Figure K.4: CIFAR-10C OOD detection AUROC results for severity level one. Across all methods, the best-performing predictive is closed-form (▬).

Table K.6: Kullback-Leibler (KL) divergence to the true predictive distributions (estimated using $10,000$ Monte Carlo samples) for different approximation methods on the ImageNet validation set using ViT Little backbones. The mean KL divergence is calculated over the validation set. Error bars represent two standard deviations over independently trained models using different seeds.

| Method | KL Divergence to True Predictive |
|---|---|
| NormCDF | $0.0057 \pm 0.0008$ |
| Sigmoid | $0.0064 \pm 0.0009$ |
| Softmax mean field | $0.0330 \pm 0.0042$ |
| Softmax Laplace bridge | $1.7900 \pm 0.1254$ |

## K.6  BCE Loss Performance Gains

Table K.7 shows that the Softmax and Sigmoid activation functions (and corresponding losses) already show improved performance on the *vanilla models* compared to softmax, highlighting that the performance gains we observe cannot only be attributed to our closed-form predictive approximations but also the favorable training dynamics of the class-wise BCE loss.

Table K.7: NLL results on ImageNet using ResNet-50 backbones and different activation functions. Error bars represent two standard deviations.

| Activation Function | NLL |
|---|---|
| NormCDF | $0.9345 \pm 0.0072$ |
| Softmax | $0.9369 \pm 0.0025$ |
| Sigmoid | $0.9146 \pm 0.0046$ |

## K.7  Further Out-of-Distribution Detection Results

Fig. K.5 shows OOD detection results across all ImageNet-C severity levels. Our closed-form predictives consistently outperform Softmax.

## K.8  Comparison to Evidential Deep Learning Methods

Table K.8 provides results for two evidential deep learning methods, EDL [46] and PostNet [6]. (We adopt the naming convention of [40].) We make the comparison because these methods also yield closed-form predictives and second-order quantities of interest, as they build Dirichlet distributions whose advantages we discuss in Section 4.1. Following our experimental setup (Section 6), we optimise the hyperparameters of both methods using a ten-step Bayesian Optimization scheme in Weights & Biases. Compared with the results presented in the main text, our approach outperforms the two EDL methods across all metrics, significantly so on NLL and OOD AUROC. The results on the latter metric indicate that our moment-matched Dirichlet distributions capture second-order epistemic information more effectively than the loss-based EDL methods, which aligns with the findings of Bengs et al. [4].

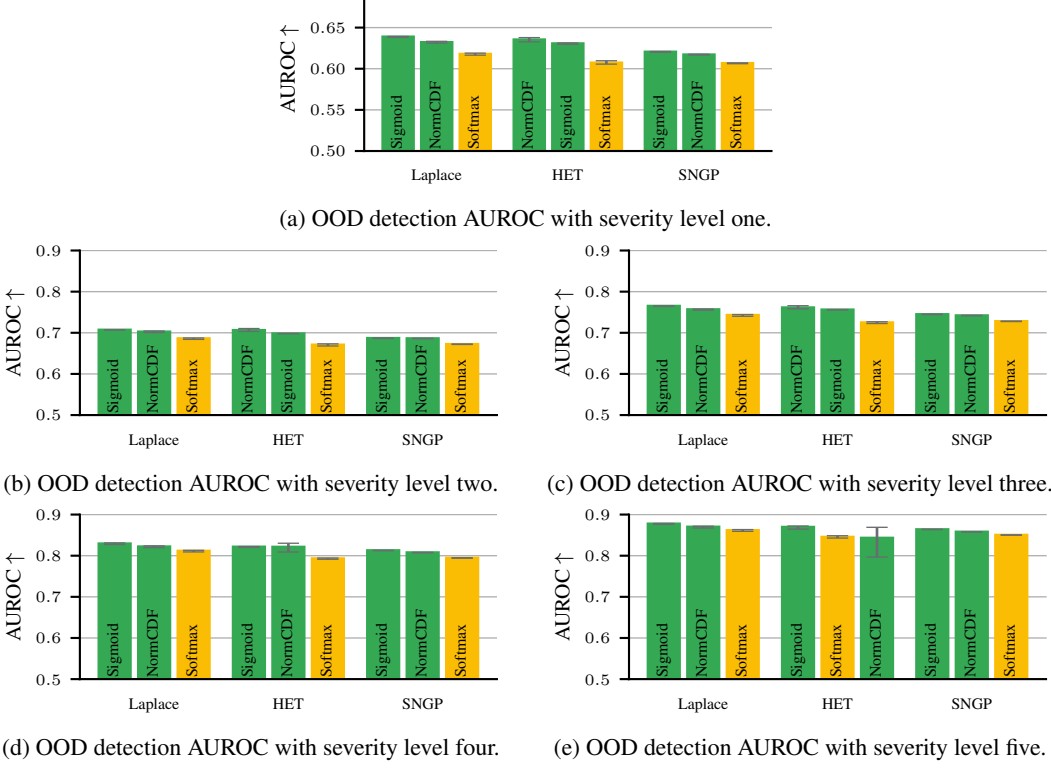

(a) OOD detection AUROC with severity level one.

(b) OOD detection AUROC with severity level two.

(c) OOD detection AUROC with severity level three.

(d) OOD detection AUROC with severity level four.

(e) OOD detection AUROC with severity level five.

Figure K.5: The OOD detection performance of all methods increases steadily as we increase the severity of the perturbed half of the mixed dataset on the ImageNet validation dataset. Our closed-form predictives consistently outperform Softmax.

Table K.8: Evaluation results of PostNet and EDL across CIFAR-10, CIFAR-100, and ImageNet. Values are mean $\pm$ two standard deviations.

| Dataset | Metric | PostNet | EDL |
|---|---|---|---|
| **ImageNet** | | | |
| | NLL | $1.10202 \pm 0.052$ | $1.25544 \pm 0.076$ |
| | Accuracy | $75.602 \pm 0.627$ | $75.632 \pm 0.326$ |
| | ECE | $0.058168 \pm 0.002$ | $0.033219 \pm 0.001$ |
| | OOD AUROC | $0.60675 \pm 0.013$ | $0.60167 \pm 0.018$ |
| **CIFAR-100** | | | |
| | NLL | $1.17973 \pm 0.081$ | $1.3645 \pm 0.107$ |
| | Accuracy | $75.183 \pm 0.192$ | $75.383 \pm 0.273$ |
| | ECE | $0.12295 \pm 0.012$ | $0.088213 \pm 0.005$ |
| | OOD AUROC | $0.59152 \pm 0.017$ | $0.59471 \pm 0.02$ |
| **CIFAR-10** | | | |
| | NLL | $0.37406 \pm 0.013$ | $0.40403 \pm 0.023$ |
| | Accuracy | $91.757 \pm 0.021$ | $90.274 \pm 0.043$ |
| | ECE | $0.046192 \pm 0.002$ | $0.041158 \pm 0.002$ |
| | OOD AUROC | $0.5931 \pm 0.014$ | $0.60808 \pm 0.006$ |

