# OpenReview forum: "Rethinking Approximate Gaussian Inference in Classification"
_NeurIPS.cc/2025/Conference — NeurIPS 2025 poster_

### Official Review · Reviewer_uK2i · 2025-06-11

**Clarity:** 2
**Significance:** 2
**Originality:** 3
**Rating:** 4
**Confidence:** 3

**Summary:**

The paper addresses the problem of uncertainty quantification in classification tasks. Building upon previous work on Gaussian inference in logit space, the Authors propose multiple enhancements to the standard procedure, aiming at
- Sidestepping MC sampling, via closed-form evaluations of the Gaussian integral and the use of tractable approximations of the pushforwards.
- Generalizing the final Softmax activation to sigmoid and normCDF, enabling accurate and MC-free approximations.

These methods are tested on standard image classification tasks.

**Questions:**

In line 173 they write "For example, on a ResNet-50, sampling one thousand logit vectors from the logit-space Gaussian means an approximately 7% overhead on the forward pass". Where is this number coming from?

**Ethical Concerns:**

["NO or VERY MINOR ethics concerns only"]

**Final Justification:**

My final thoughts after the rebuttal phase are positive. I hope that the explanations provided, which offer much better context for the methodology, will be included in the revised manuscript by the Authors. Also, the reported timings now make this a very compelling method.

I maintain my endorsement for publication, but due to a lack of expertise in this specific area, I am not comfortable increasing further my already positive score.

**Limitations:**

yes

**Quality:**

3

**Strengths And Weaknesses:**

## Strengths

- The proposed methods are principled, and they address the concrete problem of MC sampling for approximate Gaussian inference.
- There are some theoretical analyses supporting the proposed methodology
- Experimental results tested a good variety of activations and methods on standard image classification benchmarks, showcasing reasonable performance.

## Weaknesses

- From a readability perspective, I think that the paper could be greatly improved. While there are some extensive appendices and many details are reported, while reading the paper, I had a hard time piecing together the information and getting an overall picture. Some concrete examples:
    - One of the main ingredients of the recipe is the approximation in equations 10 and 12, which is introduced unceremoniously by a simple $\approx$ sign, without any further explanation highlighting its importance.
    - A "related works" or similar section is missing, and I find the broader literature around the problem not adequately introduced. The works that are discussed are instrumental in the description of the methodology, but a broader context would be appreciated.
    - Sections 3 (and its subsections) and 4 describe an undifferentiated sequence of additional methods for which it is hard (at least to me) to define a hierarchy of importance, and no indication is given about their possible use in concrete scenarios.
    - As reflected in my confidence score, I do not consider myself an expert in uncertainty quantification. That being said, I had some hard times understanding what key metrics were evaluated in the experiments. For most of them, the inexperienced reader is diverted to external references to get some intuition of the tested metric.
- I think that the experimental section critically lacks a report on the timings. One of the motivations behind the proposed methods is computational efficiency, which is theoretically discussed in Section 5. One of the first things I was looking for while reading the experimental section is a report of the actual timings to compute the predictive for the various methods, but I couldn't find one.

---

> ### Author Rebuttal · Authors · 2025-07-31
>
> We are grateful to the reviewer for their careful evaluation of our paper. We thank them for acknowledging the principled nature of our method, the theoretical analyses that support our claims, and the extensive set of experiments.
>
> **Eqs. (10) and (12).** We agree that the approximation in Eqs. (10) and (12) is the key approximation in our work. Therefore, it should be better introduced. It consists of bringing the expectation inside the fraction: the expectation of the fraction is approximated by the fraction of the expectation. Such an approximation is sometimes referred to as a mean-field approximation [1], although in this work, we specifically use the term to describe a method proposed for approximating the Gaussian softmax integral [2]. We discuss the quality of this approximation later in the paper, both theoretically (Theorem 2.1) and empirically (in the synthetic experiment, Figure 1). As the variance across the different classes decreases, the approximation becomes more accurate (in the limiting deterministic case, the approximation is exact). The aim of Theorem 2.1 is to quantify this limiting behaviour.
>
> To improve the readability of the paper, we propose to explain in words after Eqs. (10) and (12) what the approximation consists of, and point the reader toward Theorem 2.1 to give the reader a glimpse of the analysis that is to come.
>
> **Related work.** A Related Work section indeed greatly helps contextualise the present work in the literature. To avoid repetition, we have included the full text of the new Related Work section in our response to Reviewer `LBev`.
>
> **Sections 3 and 4.** Section 3 addresses how to obtain Dirichlet distributions, which offer more expressivity than a single predictive value. These can be used, for instance, for uncertainty disentanglement or out-of-distribution (OOD) detection, as we showcase in the experiments.
>
> Section 4 discusses the advantageous training opportunities that the element-wise sigmoid and normCDF activations provide. The goal is to demonstrate that these activations are not only beneficial for approximate Gaussian inference but also for training the underlying neural network.
>
> We appreciate the reviewer's point about connecting these sections to the broader framework. To address this, we have expanded our introduction with a detailed list of contributions that explicitly shows how each section builds toward our overall framework. We have pasted it in full in our response to Reviewer `LBev`.
>
> **Used metrics.** While Appendix H provides a formal overview of all the metrics we use, we agree with the reviewer that a more intuitive description would aid readability. We provide such an overview below and also give more context in the updated paper.
>   - The **negative log-likelihood (NLL)** metric measures how well the trained model fits the test datasets (i.e., whether it gives a high probability to the correct label). Given the model's predicted probability $p(y \mid x)$ of label $y$ for input $x$, we compute $- \log p(y \mid x)$. As the negative of the log probability scoring rule, NLL penalises both over- and underconfidence, reaching its minimum only when the model outputs the ground-truth class probabilities. Lower values indicate better performance (indicated by the arrows in each plot's $y$-axis).
>   - The **expected calibration error (ECE)** measures whether a model's confidence aligns with its actual accuracy. We define confidence as the maximum predicted class probability (e.g., for predicted probabilities $(0.3, 0.2, 0.5)$, the confidence is $0.5$). Test samples are binned into fifteen equal intervals over $[0, 1]$ based on their confidence scores. Within each bin, we compute the L1 distance between the average confidence and the actual accuracy, then average this distance across all bins. A well-calibrated model has similar confidence and accuracy in each bin.
>   - The **correctness prediction task** is a binary problem to determine whether the model can predict its own correctness. For each input $x$, the model's predicted class $\hat{y}(x)$ is the argmax of the predicted probability vector. We define the correctness indicator as $\ell(x) := 1[\hat{y}(x) = y]$, which is one if and only if the model correctly predicts the label for input $x$. These serve as the binary labels for the correctness prediction task. Given the model's uncertainty estimate $u(x) \in [0, 1]$ for each input $x$ (e.g., its confidence, defined in the previous point), we evaluate how well $u(x)$ predicts $\ell(x)$ using the binary cross-entropy loss over a test dataset. The negative of this value is often referred to as the log probability strictly proper scoring rule.
>   - Like correctness prediction, the **OOD detection problem** is also a binary prediction task, where a mixture of in-distribution and out-of-distribution samples is provided. The model's uncertainty estimate $u(x)$ (see above) is tasked to separate in-distribution and out-of-distribution samples. To measure the model's performance, we use the Area Under the Receiver Operating Curve (AUROC) metric, which measures how separable the in-distribution and out-of-distribution samples are based on the model's uncertainty estimate (where 1.0 indicates perfect separation and 0.5 indicates random performance).
>
> **Timings.** We agree that including more reports of timings in the experimental section helps to underscore the efficiency of our method. Regarding your specific question about the 7% overhead, this measurement was from preliminary experiments on different hardware (MacBook Pro M4). For more systematic comparisons, we measured the time to obtain the predictives from the logit-space Gaussians compared to the time of a forward pass on an NVIDIA RTX 2080 Ti GPU (with similar results on Tesla A100 GPUs). As shown below, ResNet-50 with 1000 samples shows 6-8% overhead depending on the batch size, confirming our claim.  In contrast, the runtime cost of computing our closed-form predictives is negligible compared to a forward pass, especially for larger batch sizes.
>
> **ResNet-26 (C=10) Runtime Overhead (sampling time as % of total inference time)**
>
> | Batch Size | Mean-Field | 1 sample | 10 samples | 100 samples | 1000 samples |
> |------------|------------|-----------|------------|-------------|--------------|
> | BS=1       | 2.75%      | 9.82%     | 10.63%     | 10.70%      | 10.70%       |
> | BS=16      | 2.04%      | 10.61%    | 11.51%     | 11.51%      | 11.92%       |
> | BS=32      | 2.11%      | 10.98%    | 11.94%     | 11.96%      | 12.42%       |
> | BS=64      | 2.11%      | 10.88%    | 11.90%     | 11.95%      | 12.34%       |
> | BS=128     | 2.04%      | 8.12%     | 8.84%      | 8.96%       | 9.61%        |
> | BS=256     | 1.17%      | 4.31%     | 4.73%      | 4.93%       | 7.14%        |
>
> **Wide-ResNet-26-5 (C=100) Runtime Overhead (sampling time as % of total inference time)**
>
> | Batch Size | Mean-Field | 1 sample | 10 samples | 100 samples | 1000 samples |
> |------------|------------|-----------|------------|-------------|--------------|
> | BS=1       | 1.29%      | 8.54%     | 9.36%      | 9.40%       | 9.28%        |
> | BS=16      | 1.16%      | 5.94%     | 6.58%      | 6.84%       | 8.56%        |
> | BS=32      | 0.86%      | 3.36%     | 3.76%      | 3.98%       | 6.95%        |
> | BS=64      | 0.47%      | 1.83%     | 2.03%      | 2.12%       | 5.82%        |
> | BS=128     | 0.23%      | 0.90%     | 1.04%      | 1.04%       | 4.88%        |
> | BS=256     | 0.12%      | 0.46%     | 0.53%      | 0.84%       | 4.47%        |
>
> **ResNet-50 (C=1000) Runtime Overhead (sampling time as % of total inference time)**
>
> | Batch Size | Mean-Field | 1 sample | 10 samples | 100 samples | 1000 samples |
> |------------|------------|-----------|------------|-------------|--------------|
> | BS=1       | 1.15%      | 5.43%     | 5.88%      | 6.09%       | 6.30%        |
> | BS=16      | 0.32%      | 1.25%     | 1.43%      | 1.70%       | 8.04%        |
> | BS=32      | 0.16%      | 0.64%     | 0.72%      | 1.34%       | 7.51%        |
> | BS=64      | 0.09%      | 0.36%     | 0.41%      | 1.13%       | 7.75%        |
> | BS=128     | 0.05%      | 0.20%     | 0.22%      | 1.00%       | 7.82%        |
> | BS=256     | 0.03%      | 0.10%     | 0.18%      | 0.95%       | 8.05%        |
>
> These results demonstrate that even with a diagonal covariance structure, MC sampling yields a nontrivial overhead, especially for large numbers of classes and samples. In comparison, our closed-form approach has negligible computational cost.
>
> Thank you again for your constructive feedback. We have incorporated all your suggestions to improve the paper's readability: adding a Related Work section, a list of contributions to explain the organisation of the paper, clear explanations of the key approximations, timing results, and intuitive descriptions of all metrics. We believe these changes significantly strengthen the presentation of our work.
>
> [1]: Friedman & Koller, (2009). Probabilistic Graphical Models -- Principles and Techniques
>
> [2]: Lu et al., (2021). Mean-field approximation to Gaussian-Softmax integral with application to uncertainty estimation

---

> > ### Comment · Reviewer_uK2i · 2025-08-04
> >
> > I thank the Authors for their thorough response.
> > I hope that the explanations provided, which offer much better context for the methodology, will be included in the revised manuscript. Also, the reported timings now make this a very compelling method.
> >
> > I maintain my endorsement for publication, but due to a lack of expertise in this specific area, I am not comfortable increasing further my already positive score.

---

### Official Review · Reviewer_a52w · 2025-06-24

**Clarity:** 3
**Significance:** 2
**Originality:** 3
**Rating:** 4
**Confidence:** 4

**Summary:**

The paper proposes an alternative to Monte Carlo sampling when computing the predictive distribution in Bayesian NNs which model the logits using a Gaussian distribution per class in classification tasks.  The key idea is to decouple the nominator and denominator when normalizing the logits and compute each term using closed-form formulas involving Gaussian densities. The authors present a bound on the error invoked by the approximations made to derive the predictive distribution. The authors further fit a Dirichlet distribution over the simplex by means of moment matching. Empirically, the authors show comparable results in terms of accuracy to standard MC-based predictive distribution calculation while benefiting from increased uncertainty estimation and OOD detection.

**Questions:**

See Strengths And Weaknesses section.

**Ethical Concerns:**

["NO or VERY MINOR ethics concerns only"]

**Final Justification:**

Please see my comment to the review.

**Limitations:**

Yes.

**Paper Formatting Concerns:**

No paper formatting concerns.

**Quality:**

2

**Strengths And Weaknesses:**

Strengths:
- The paper deals with a problem that is given less attention in the literature as far as I am aware. Commonly MC samples are taken which can be suboptimal.
- Most steps in the method are supported either empirically or theoretically, for instance estimating the expectation of a ratio with the ratio between the expectation of the nominator and the expectation of the denominator. Or, using Beta distribution for relevant activation functions.
- The paper is written clearly and the method has novelties in it.
- The empirical results are reasonable, the method is comparable to the Softmax baseline in terms of accuracy and tends to improve in other metrics.

Weaknesses/Questions:
- The scope of the method has limitations as it assumes a Gaussian distribution over the logits per class which can be overly restrictive.
- The authors present the computational benefit as one of the advantages of their approach. Yet, besides section 5 which presents it in terms of the number of operations I didn't notice any discussion. Perhaps the authors can show a comparison in terms of wall-clock time and memory to understand better when their method is preferred (e.g., as a function of the number of MC samples and classes).
- Perhaps I missed it, but do you have results for using your approximation for pre-trained networks with common cross-entropy loss and Softmax activation? If indeed models need to be trained or fine-tuned to accommodate the objective in Eq. 29 it can be a harsh restriction, especially in this era of large models.
- In light of the Gaussian assumption over the logits, I believe that authors should have discussed, and compared where appropriate, to other alternatives for the Softmax likelihood. Several examples can be found in the Gaussian process literature. Non exhaustive list: one-vs-each [1], robust-max [2], Dirichlet observation model [3], logistic-softmax [4],  stick-breaking [5], and GP-Tree [6].
- Technical questions:
  - In Eq. 43, why do you assume that the expectation of $Q_c$ is equal for all classes?
  - In Eq. 58, second line, shouldn't the last term be $\sum_{c=1}^C p_c \frac{(p_c - \tilde{p}_c)^2}{2 \omega (\tilde{p}_c)^2)}$?
  - Is there a justification to assume that $Var(\sum_{c=1}^C Q_c) \rightarrow 0$?
- Minor:
  - In Eq. 6 should it be $p(f(x))$ in the last term?
  - Broken link in line 876 at the Appendix.

[1] AUEB, T. R. (2016). One-vs-each approximation to softmax for scalable estimation of probabilities. Advances in Neural Information Processing Systems, 29.
[2]  Hernández-Lobato, D., Hernández-Lobato, J., & Dupont, P. (2011). Robust multi-class Gaussian process classification. Advances in neural information processing systems, 24.
[3] Kapoor, S., Maddox, W. J., Izmailov, P., & Wilson, A. G. (2022). On uncertainty, tempering, and data augmentation in Bayesian classification. Advances in neural information processing systems, 35, 18211-18225.
[4] Galy-Fajou, T., Wenzel, F., Donner, C., & Opper, M. (2020, August). Multi-class Gaussian process classification made conjugate: Efficient inference via data augmentation. In Uncertainty in artificial intelligence (pp. 755-765). PMLR.
[5] Linderman, S., Johnson, M. J., & Adams, R. P. (2015). Dependent multinomial models made easy: Stick-breaking with the Pólya-Gamma augmentation. Advances in neural information processing systems, 28.
[6] Achituve, I., Navon, A., Yemini, Y., Chechik, G., & Fetaya, E. (2021, July). GP-Tree: A Gaussian process classifier for few-shot incremental learning. In International conference on machine learning (pp. 54-65). PMLR.

---

> ### Author Rebuttal · Authors · 2025-07-31
>
> We are very grateful to the reviewer for their thorough read of our paper, including the technical appendices, and for acknowledging the novelty of our work, the clarity of the presentation, the empirical and theoretical arguments that support the claims made in the paper, and the extensive set of experiments.
>
> Below, we address the points raised by the reviewer one by one.
>
> **Logit-space Gaussian assumption.** As the reviewer points out, our work concerns methods that output Gaussian distributions over the logits. While this setting might seem restrictive at first, it encompasses many state-of-the-art methods for uncertainty quantification, making it arguably one of the dominant paradigms in uncertainty quantification for deep learning. Our empirical comparison of approximate Gaussian inference methods against evidential deep learning methods also supports this claim (see our response to Reviewer `LBev`). In the revised manuscript, we include a Related Work section which, among other things, describes the significance of such methods. To avoid repetition, we have pasted this new related work section in full in our response to Reviewer `LBev`.
>
> **Computational benefits.** We agree that providing more timing reports in the experimental section emphasizes the benefits of sample-free inference. We compare the time to construct Gaussians with diagonal covariance compared to the time of a forward pass. Below we show results on an NVIDIA RTX 2080 Ti. NVIDIA Tesla A100 GPU gave similar results. In reality, the overhead of MC sampling may be even greater, as each of these logit Gaussian samples needs to be mapped through the softmax function. In contrast, the runtime cost of computing our closed-form predictives is negligible compared to a forward pass.
>
> **Resnet-26 (C=10)**
>
> | Batch Size | 1 samples | 10 samples | 100 samples | 1000 samples |
> |------------|-----------|------------|-------------|--------------|
> | BS=1 | 10.47% | 11.33% | 11.32% | 11.09% |
> | BS=16 | 11.26% | 12.09% | 12.09% | 12.35% |
> | BS=32 | 11.42% | 12.47% | 12.47% | 12.75% |
> | BS=64 | 11.52% | 12.51% | 12.51% | 12.77% |
> | BS=128 | 8.91% | 9.65% | 9.71% | 9.94% |
> | BS=256 | 4.88% | 5.29% | 5.38% | 7.02% |
>
> **Wide-Resnet-26-5 (C=100)**
>
> | Batch Size | 1 samples | 10 samples | 100 samples | 1000 samples |
> |------------|-----------|------------|-------------|--------------|
> | BS=1 | 10.45% | 11.39% | 11.28% | 11.41% |
> | BS=16 | 7.65% | 8.32% | 8.50% | 9.21% |
> | BS=32 | 4.55% | 4.90% | 5.07% | 7.19% |
> | BS=64 | 2.68% | 2.88% | 2.99% | 5.72% |
> | BS=128 | 1.62% | 1.72% | 1.78% | 4.83% |
> | BS=256 | 1.05% | 1.12% | 1.36% | 4.32% |
>
> **Resnet-50 (C=1000)**
>
> | Batch Size | 1 samples | 10 samples | 100 samples | 1000 samples |
> |------------|-----------|------------|-------------|--------------|
> | BS=1 | 6.37% | 6.87%      | 6.97%       | 5.94%        |
> | BS=16 | 2.32% | 2.51%      | 2.69%       | 7.24%       |
> | BS=32 | 1.58% | 1.69%      | 2.22%       | 6.86%        |
> | BS=64 | 1.21% | 1.26%      | 1.72%       | 6.79%        |
> | BS=128 | 0.88% | 0.92%      | 1.40%       | 6.26%        |
> | BS=256 | 0.69%  | 0.73%      | 1.12%       | 5.88%        |
>
> **Pre-trained models.** Thank you for raising the important point about pre-trained models. While Appendix K.1 discusses the limitations of applying our approximation directly to off-the-shelf softmax models, we have an encouraging observation from our experiments: In our ImageNet setup, we started with pre-trained ResNet-50 and ViT-Little backbones and fine-tuned them with sigmoid/normCDF activations. We observed that these models adapted remarkably quickly, achieving near-optimal loss and accuracy after just 4-5 ImageNet epochs. This suggests that our approach remains practical in the era of large models, requiring only brief fine-tuning rather than training from scratch. Given the strong uncertainty quantification benefits demonstrated throughout the paper, we believe this short adaptation period is a worthwhile trade-off. This finding also opens interesting directions for future work on efficient uncertainty transfer between pre-trained models and our framework.
>
> **Softmax likelihood alternatives.** We appreciate the pointer to alternatives to the softmax likelihood and have clarified this in the paper. Of the cited works, robust-max [1], logistic-softmax [2], stick-breaking [3], and GP-Tree [4] do replace the softmax likelihood, whereas one-vs-each [5] is a training surrogate for softmax (not a normalised likelihood) and the Dirichlet/tempered perspective for BNNs [6] retains softmax while re-specifying aleatoric confidence. All of these approaches require sampling or one-dimensional quadrature at inference. By contrast, given Gaussian logits from any back-end, our method provides closed-form, sample-free predictives and second-order Dirichlet quantities, with a theoretical error analysis. To better contextualise our work in this significant body of literature, we now include a Related Work section in the paper. To avoid repetition, we have pasted it in full in our response to Reviewer `LBev`.
>
> ### Technical Questions
>
> **T1:** Appendix C1 performs an informal theoretical analysis specifically under the setting of the synthetic experiment from Figure 1. In Eq. (43), we use that the logit means and variances are sampled i.i.d. for each class; hence, $\mathbb{E}[Q_c] \\approx \\mathbb{E}[Q_1]$ for all $c$, and they are of the same order in the number of classes. The reviewer is correct in saying that this is not exact, but an approximation. Thank you for spotting this. We have replaced the equality sign with an approximation sign and explained the source of this approximation. Note that this approximation is valid in this informal analysis because we only care about the order of the quantities as a function of $C$, and in this case, $\\mathbb{E}[Q_c]$ will be the same order as $\\mathbb{E}[Q_1]$, i.e., constant in $C$.
>
> **T2:** Thank you for catching the missing summation in Eq. (58). We have now corrected this, and the rest of the analysis carries through as before. For completeness, we demonstrate this here:
>
> $$= \sum_{c=1}^C p_c\left(\frac{p_c - \tilde p_c}{p_c} + \frac{(p_c-\tilde p_c)^2}{2\omega_c(\tilde p_c)^2}\right)$$
>
> $$= \\sum_{c=1}^C p_c - \\sum_{c=1}^C \\tilde p_c + \\sum_{c=1}^Cp_c \\frac{(p_c-\\tilde p_c)^2}{2\\omega_c(\\tilde p_c)^2}$$
>
> $$= \sum_{c=1}^C\frac{p_c}{2\omega_c(\tilde p_c)^2}\cdot O\left(\operatorname{Var}\left(\sum_{c'=1}^C Q_{c'}\right)^{1/2}\right)^2$$
>
> $$= O\left(\operatorname{Var}\left(\sum_{c=1}^CQ_c\right)\right)$$
>
> as $\operatorname{Var}\left(\sum_{c=1}^C Q_c\right)\to 0$, where $\omega_c(\tilde p_c)\in [p_c,\tilde p_c] \text{ or }[\tilde p_c,p_c]$, where we used that $p_c$, $\tilde p_c$, and hence $\omega_c(\tilde p_c)$ is bounded away from $0$ by compactness of $\mathcal K$.
>
> **T3:** In practice we observe that $\operatorname{Var}\left(\sum_{c=1}^C Q_c\right)$ is very small (between $0.001$ and $0.01$). Meanwhile, the individual variances $\operatorname{Var}(Q_c)$ remain meaningful for uncertainty quantification, as we see in the various experiments that the approximate Gaussian inference methods significantly outperform the neural networks' point estimates (Appendix K.5 for NLL). Moreover, there is a bound of the form $\operatorname{Var}\left(\sum_{c=1}^C Q_c\right) \leq C \sum_{c=1}^C \operatorname{Var}(Q_c)$. On the other hand, $\sum_{c=1}^C \operatorname{Var}(Q_c)$ cannot be bounded by $\operatorname{Var}\left(\sum_{c=1}^C Q_c\right)$ in general. This shows that, for our approximation to be good, we do not necessarily require the individual uncertainties in the classes to be small, a small $\operatorname{Var}\left(\sum_{c=1}^C Q_c\right)$ is sufficient.
>
> To demonstrate this, we run a synthetic experiment where we sample $200$ logit means and covariance matrices such that $\sum_{c=1}^C \operatorname{Var}(Q_c) > 0.1 \cdot C$, take the minimum $\operatorname{Var}\left(\sum_{c=1}^C Q_c\right)$, and show that it is much smaller than $\sum_{c=1}^C \operatorname{Var}(Q_c)$, see the table below. This provides an existence proof that our approximation can work well even with significant individual class uncertainties, as long as  $\operatorname{Var}\left(\sum_{c=1}^C Q_c\right)$ is small.
>
> | C | 2 | 3 | 4 | 5 | 6 | 8 | 10 | 15 | 20 |
> | ---------------------------- | -------- | -------- | -------- | -------- | -------- | -------- | -------- | -------- | -------- |
> | Min Var[Sum sigmoid(Q_c)]    | 0.003678 | 0.005642 | 0.006437 | 0.020943 | 0.013800 | 0.029320 | 0.062187 | 0.034957 | 0.058531 |
> | Sum Var[sigmoid(Q_c)] at min | 0.430854 | 0.307303 | 0.604298 | 0.622324 | 0.986399 | 1.154063 | 1.582350 | 1.998524 | 2.906889 |
>
> ### Minor
>
> **M1:** Our $\mathrm f(x)$ is a Gaussian distribution, not a point estimate (see Eq. 5). So we believe our Eq. (6) is correct.
>
> **M2:** We have now fixed the link in the appendix. Thank you for pointing it out.
>
> Thank you again for your detailed review. We believe we have addressed your main concerns regarding the scope of our method, computational benefits, and technical questions. We are happy to clarify any remaining points.
>
> [1]: Hernández-Lobato et al., (2011). Robust Multi-Class Gaussian Process Classification
>
> [2]: Galy-Fajou et al., (2020). Multi-Class Gaussian Process Classification Made Conjugate: Efficient Inference via Data Augmentation
>
> [3]: Linderman et al., (2015). Dependent Multinomial Models Made Easy: Stick-Breaking with the Pólya-gamma Augmentation
>
> [4]: Achituve et al., (2021). GP-Tree: A Gaussian Process Classifier for Few-Shot Incremental Learning
>
> [5]: Titsias, (2016). One-vs-Each Approximation to Softmax for Scalable Estimation of Probabilities
>
> [6]: Kapoor et al., (2022). On Uncertainty, Tempering, and Data Augmentation in Bayesian Classification

---

> > ### Comment · Reviewer_a52w · 2025-08-04
> >
> > I would like to thank the authors for their responses, which addressed both my concerns and those of my fellow reviewers. Regarding the results: while I acknowledge that the proposed method improves upon standard Monte Carlo sampling, the gains in both run-time and uncertainty metrics appear to be moderate in my view (particularly when using the Laplace approximation). Moreover, considering that the method still requires training over several epochs for a pre-trained network, I find that—without diminishing the novelty and contributions of the work—its potential impact is limited.
> > I have a few follow-up questions I would appreciate clarification on:
> > 1. I am curious why $w_c(\tilde{p}_c)$ in the denominator can be treated as a constant. Is there anything that prevents it from becoming arbitrarily close to zero (if $p_c \rightarrow 0$)?
> > 2. Just to confirm, the assumption that $V(\sum_{c=1}^C Q_c) \rightarrow 0$  is primarily based on empirical observations?
> >
> > With these caveats, I have decided to raise my score to 4, but not beyond.

---

> > > ### Author Response · Authors · 2025-08-04
> > >
> > > We thank the reviewer for their response and for raising their score. We agree that, for pre-trained networks with a softmax activation, the method requires fine tuning over a few epochs. However, we also demonstrate that there are no caveats to training with the sigmoid or normCDF BCE loss. On the contrary, as demonstrated in both the present and previous works [1, 2], they outperform the softmax CE loss for classification (significantly so on ECE and OOD detection) and at the same time eliminate the runtime overhead of MC sampling. Hence we recommend that uncertainty quantification practitioners choose these activations and losses for classification instead. (In addition, we note that the evaluated HET and SNGP methods fundamentally require fine-tuning the pre-trained deterministic backbone, meaning our proposed activations yield _no overhead_ over softmax. For the Laplace approximation on pre-trained softmax backbones, there is a net overhead of four-five epochs of fine-tuning.)
> > >
> > > 1. We agree that this is non-trivial, and we now clarify this in the revised manuscript. $p_c$ and $\tilde p_c$ are both continuous functions of $\mu_c$ and $\sigma^2_c$, so write $p_c = p_c(\mu_c,\sigma^2_c)$, $\tilde p_c = \tilde p_c(\mu_c,\sigma^2_c)$. We assumed $(\mu_c, \sigma^2_c) \in \mathcal K \subset \mathbb R\times \mathbb R_{\geq 0}$ where $\mathcal K$ is compact. $\omega_c(\tilde p_c)$ is not a constant, in that it depends on $\tilde p_c$, and hence on $\mu_c$ and $\sigma^2_c$. For the derivation to hold, what we need to show is that it is bounded away from $0$. Write $m:= \inf_{(\mu_c,\sigma^2_c)\in \mathcal K} p(\mu_c,\sigma^2_c).$ Since $\mathcal K$ is compact and $p$ is a continuous function, $m$ is attained, i.e., there are $(\hat \mu_c, \hat \sigma^2_c) \in \mathcal K$ such that $m = p(\hat \mu_c, \hat \sigma^2_c)$. But $p(\mu_c,\sigma^2_c) > 0$ for all $(\mu_c, \sigma^2_c) \in  \mathbb R\times \mathbb R_{\geq 0}$. So $m>0$. Similarly, letting $\tilde m:= \inf_{(\mu_c,\sigma^2_c)\in \mathcal K} \tilde p(\mu_c,\sigma^2_c)$, $\tilde m$ is attained so $\tilde m>0$. Now, for $(\mu_c, \sigma^2_c) \in \mathcal K$, $\omega_c(\tilde p_c) \in [p_c,\tilde p_c] \text{ or } [\tilde p_c, p_c]$, so $\omega_c(\tilde p_c) > \min (m, \tilde m) >0$.
> > > 2. For the error analysis, it is natural to consider the limit $V\left(\sum_{c=1}^C Q_c\right) \to 0$, as our approximation becomes exact there. The _practical relevance_ of Thm. 2.1 is indeed highlighted by empirical observations: (i) the quantity being within $[0.001, 0.01]$ in all our experiments on ImageNet, CIFAR-10, and CIFAR-100; and (ii) the observation that a small $V\left(\sum_{c=1}^C Q_c\right)$ does not prevent the _individual variances_ $V(Q_c)$ from being orders of magnitude larger, as shown in our rebuttal.
> > >
> > > **References:**
> > >
> > > ---
> > >
> > > [1]: Wightman et al., (2021). ResNet strikes back
> > >
> > > [2]: Li et al., (2025). Binary Cross-Entropy Loss Leads to Better Calibration than Focal Loss on Imbalanced Multi-Class Datasets

---

> > > > ### Comment · Reviewer_a52w · 2025-08-04
> > > >
> > > > Thank you for the clarifications, I maintain the current score of 4 following the rebuttal.

---

### Official Review · Reviewer_LBev · 2025-06-26

**Clarity:** 4
**Significance:** 2
**Originality:** 2
**Rating:** 4
**Confidence:** 5

**Summary:**

The authors use the fact that expectations over one-dimensional Gaussian distributed random variables
are analytically tractable (or can be tightly approximated) for a number of transformations,
such as the exponential function, or normal cdf, which often appear in classification objectives.
They use this to improve classification performance by predicting Gaussians in the logit space
and transforming them using this result for a richer training signal.

The method is evaluated with several architectural variations on a range of image classification data sets
and performance metrics, that demonstrate not only improved predictive performance, but also
improved calibration.

**Questions:**

Given the points mentioned above:
- Q1: Can the authors provide a deeper discussion that highlights the strengths and innovations of their approach with respect to prior probit-based knowledge in the literature?
- Q2: Can the authors discuss their approach with relation to the greater context of approaches mentioned above, and, if possible time-wise, a comparison against evidential deep learning or one of the Taylor approximation-based methods?

**Ethical Concerns:**

["NO or VERY MINOR ethics concerns only"]

**Final Justification:**

I went through the other reviews as well as the rebuttals. I still think that the theoretical contribution and embedding into prior work are limited, but that the strengths are just enough overall outweigh them.

**Limitations:**

Societal implications are not discussed. I don't consider there to be any that go beyond the general implications due to deep learning. As such this is not a weakness of the paper.

**Paper Formatting Concerns:**

I have no concerns regarding the paper formatting.

**Quality:**

3

**Strengths And Weaknesses:**

## Strengths
- The paper is well-written and can be followed quite easily
- The method is properly motivated with an extensive set of experiments (however, see the weakness below)
- The proposal is generic enough to be added to a wide variety of prior approaches
- The authors provide a detailed implementation

## Weaknesses
### Significance
The closed-form probit solution to the expectation of a normally distributed random variable is well-known in the literature (for recent work see, e.g., Kristiadi et al., 2020; Haussmann et al., 2020; Li et al., 2025 among others and the references therein).
As such, I would consider this common knowledge in the field with the remaining primary contribution being the application in a multi-class setting, which is rarer.
(At least, I can't point to a specific reference, although it might have been explored.)
As such, the paper's main contribution is its detailed discussion and evaluation of such closed-form approximations.

### Prior work
The second main weakness of the paper is its lack of discussion of prior work and experimental comparison against such work. Given the small theoretical contribution, such a discussion and evaluation
would be the main strength of this work.

The two main areas of research that are (partially) omitted are

- **deterministic/sampling-free BNN methods:**
As there is a large variety of approaches in that area of research, I will only refer to a few representative ones. These are, Gast et al. (2020); Wu et al., (2019); Haussmann et al. (2020) for older approaches and Wright et al. (2024); Li et al. (2025) and the references therein for current work.
All of them aim to infer BNNs in a sampling-free manner and apart from Wright et al. also discuss sampling-free classification approaches.
Gast et al. parameterize a Dirichlet distribution (see below). Wu et al. derive a Taylor approximation-based approach (but do not evaluate it and stick to regression). Haussmann et al. discuss the Probit approximation, but stick to evaluating an adaptation of Wu et al.'s approximation. Finally, Li et al. use the Probit approximation for their classification experiments, refering to Kristiadi et al. (2020) as their motivation.
Given their relevance such related work should be discussed and compared against.

- **Dirichlet-based approaches:**
Apart from Gast et al. (2020) mentioned above, a major research direction is known as _Evidential Deep Learning_. See Sensoy et al., (2018) who introduced it and the diverse literature that builds on it.
While Sensoy et al. (2018) are referred to in passing, their relation should be discussed and compared against in greater detail.



_____
Gast et al. (2018), _Lightweight Probabilistic Deep Networks_
Haussmann et al. (2020), _Sampling-Free Variational Inference of Bayesian Neural Networks by Variance Backpropagation_
Kristiadi et al. (2020), _Being Bayesian, Even Just a Bit, Fixes Overconfidence in ReLU Networks_
Li et al. (2025), _Streamlining Prediction in Bayesian Deep Learning_
Sensoy et al. (2018), _Evidential Deep Learning to Quantify Classification Uncertainty_
Wright et al. (2024), _An Analytic Solution to Covariance Propagation in Neural Networks_

---

> ### Author Rebuttal · Authors · 2025-07-31
>
> We are grateful to the reviewer for their thorough evaluation of our manuscript and for including several important references. We thank them for praising the clarity of our work, the extensiveness of our experimental analysis, the versatility of our framework, and the detailed implementation provided. We address their two central questions/concerns below.
>
> ### Significance
>
> We agree that the probit approximation for the Gaussian sigmoid integral is well-known, and certainly not novel to our work. It dates back to [1, 2], and as the reviewer points out, is used in recent works for binary classification, such as [3]. As the reviewer writes, we generalise it to multi-class settings. Furthermore, this is the first work that provides a theoretical analysis of such an approximation, to the best of our knowledge. We also show empirically that it outperforms other approximations from the literature. We extend it further to allow for not just predictive approximations but second-order Dirichlet approximations. Finally, we leverage the advantageous structure of the activation to train with the binary cross-entropy loss. Combined with our closed-form predictives, we demonstrate the improved uncertainty quantification capabilities of this pipeline.
>
> To clarify these points, we now include a List of Contributions section at the end of the introduction:
>
> **List of Contributions**
>
> Our work extends the classical probit approximation for the Gaussian sigmoid integral in binary classification [1, 2] in the following manner:
> - Generalising it to the multi-class setting (Sections 2.1 and 2.2),
> - Analysing it theoretically (Section 2.3),
> - Analysing it empirically, and how it outperforms previous approximations (Figure 1 and Section 6),
> - Generalising it to second-order distributional Dirichlet approximations (Section 3),
> - Combining it with advantageous training (Sections 4 and 6).
>
> These contributions are detailed in our revised manuscript, which also includes expanded Related Work and experimental comparisons.
>
> ### Contextualisation
>
> We agree that contextualising our work and comparing it to the existing literature is crucial, and we have spent a significant amount of effort in this rebuttal to do so.
>
> Firstly, we compare our approach to evidential deep learning methods [4, 5] and present the results below.
>
> **ImageNet**
>
> | Metric | PostNet | Sensoy et al. |
> |--------|---------|-----|
> | NLL | 1.10202 ± 0.052 | 1.25544 ± 0.076 |
> | Accuracy | 75.602 ± 0.627 | 75.632 ± 0.326 |
> | ECE | 0.058168 ± 0.002 | 0.033219 ± 0.001 |
> | OOD AUROC | 0.60675 ± 0.013 | 0.60167 ± 0.018 |
>
> **CIFAR-100**
>
> | Metric | PostNet | Sensoy et al. |
> |--------|---------|-----|
> | NLL | 1.17973 ± 0.081 | 1.3645 ± 0.107 |
> | Accuracy | 75.183 ± 0.192 | 75.383 ± 0.273 |
> | ECE | 0.12295 ± 0.012 | 0.088213 ± 0.005 |
> | OOD AUROC | 0.59152 ± 0.017 | 0.59471 ± 0.02 |
>
> **CIFAR-10**
>
> | Metric | PostNet | Sensoy et al. |
> |--------|---------|-----|
> | NLL | 0.37406 ± 0.013 | 0.40403 ± 0.023 |
> | Accuracy | 91.757 ± 0.021 | 90.274 ± 0.043 |
> | ECE | 0.046192 ± 0.002 | 0.041158 ± 0.002 |
> | OOD AUROC | 0.5931 ± 0.014 | 0.60808 ± 0.006 |
>
> Following our experimental setup (Section 6), we optimise the hyperparameters of both methods using a ten-step Bayesian Optimization scheme in Weights & Biases. Compared to the results presented in our paper, our approach outperforms the two EDL methods across all metrics, significantly so on NLL and OOD AUROC. The results on the latter metric indicate that our moment-matched Dirichlet distributions capture second-order epistemic information more effectively than the loss-based EDL methods, which aligns with the findings of Bengs et al. [6].
>
> We also now compare to an additional softmax Gaussian integral approximation from [7].
>
> Finally, we have written a new Related Work section that contextualises our work w.r.t. all references shared by the reviewer. We paste this verbatim below.
>
> **Related Work**
>
> In uncertainty quantification, methods that output logit Gaussians, which we refer to as *approximate Gaussian inference* methods, have become one of the dominant paradigms in uncertainty quantification [3, 8, 9, 10]. In classification tasks, such methods have the downside of requiring sampling at inference to transform Gaussian distributions into predictive probability vectors.
>
> In parallel, prior work has developed ways to propagate probabilistic uncertainty through neural networks [11]. Several works develop analytic propagation of covariances up to the logits [12, 13]; our method complements these by addressing the propagation of the logit-space Gaussian distributions onto the probability simplex. Few works address this last step. The Laplace bridge [14] proposes a map from Gaussians to Dirichlets on the probability simplex. However, this approach offers no guarantees on the quality of its predictives, and it severely underperforms Monte Carlo sampling. Other approaches include the mean-field approximation to the Gaussian softmax integral [15] and the approximation of [7]. In this work, we propose approximate predictives that outperform existing closed-form predictive techniques and provide second-order Dirichlet distributions for other closed-form quantities of interest. Furthermore, this is the first work to provide a theoretical analysis of such an approximation.
>
> Other works have also proposed various activations for classification to replace the softmax. Particularly relevant is [16], which composes element-wise sigmoids with normalisation and utilises auxiliary-variable augmentation to obtain closed-form ELBO terms during training; however, its predictives require Monte Carlo estimation. Related conjugate/augmentation-based objectives exist for stick-breaking and tree-structured multi-class GPs [17, 18], and for BNNs, a tempered-likelihood/Dirichlet perspective with a factorised-Gaussian surrogate has been explored [19]. Yet, all these approaches still require sampling or numerical quadrature at inference time. [1, 2, 3] use the probit approximation for the Gaussian sigmoid integral in the binary case, which we generalise to the multiclass setting. [20] approximates the softmax by providing lower bounds for point estimate training, but replacing the softmax with these bounds during training does not yield a normalised multi-class likelihood. The binary cross-entropy loss with the element-wise sigmoid activation has been reported to outperform cross-entropy in some large-scale settings [21, 22]. Our work is the first to use this advantageous loss while normalising the output to obtain proper multi-class probabilistic models.
>
> Evidential deep learning (EDL) approaches [4, 5] offer sample-free outputs through Dirichlet distributions but have been shown to be unreliable for uncertainty estimation and disentanglement [6, 23, 24, 25]. We focus on making approximate Gaussian inference methods sample-free in a principled and performant manner; however, for completeness, we compare our approach with EDL methods in Appendix L.
>
> Finally, multiple authors have explored continuous approximations to categorical distributions, which enable gradient-based optimization for discrete stochastic models through sampling, such as the Gumbel-max trick [26, 27, 28, 29]. These methods principally optimise a single categorical distribution, while the present work is concerned with constructing probability densities over the (continuous) simplex of such categorical distributions.
>
> [1]: Spiegelhalter & Lauritzen, (1990). Sequential updating of conditional probabilities on directed graphical structures
>
> [2]: MacKay, (1992). The Evidence Framework Applied to Classification Networks
>
> [3]: Kristiadi et al., (2020). Being Bayesian, Even Just a Bit, Fixes Overconfidence in ReLU Networks
>
> [4]: Sensoy et al., (2018). Evidential Deep Learning to Quantify Classification Uncertainty
>
> [5]: Charpentier et al., (2020). Posterior Network
>
> [6]: Bengs et al., (2022). Pitfalls of Epistemic Uncertainty Quantification through Loss Minimisation
>
> [7]: Shekhovtsov & Flach, (2018). Feed-forward Propagation in Probabilistic Neural Networks
>
> [8]: Collier et al., (2021). Correlated Input-Dependent Label Noise in Large-Scale Image Classification
>
> [9]: Daxberger et al., (2021). Laplace Redux
>
> [10]: Liu et al., (2020). Simple and Principled Uncertainty Estimation with Deterministic Deep Learning
>
> [11]: Gast & Roth, (2018). Lightweight Probabilistic Deep Networks
>
> [12]: Wright et al., (2024). An Analytic Solution to Covariance Propagation in Neural Networks
>
> [13]: Li et al., (2024). Streamlining Prediction in Bayesian Deep Learning
>
> [14]: Hobbhahn et al., (2022). Fast Predictive Uncertainty for Classification with Bayesian Deep Networks
>
> [15]: Lu et al., (2021). Mean-field approximation to Gaussian-Softmax integral
>
> [16]: Galy-Fajou et al., (2020). Multi-Class Gaussian Process Classification Made Conjugate
>
> [17]: Linderman et al., (2015). Dependent Multinomial Models Made Easy
>
> [18]: Achituve et al., (2021). GP-Tree
>
> [19]: Kapoor et al., (2022). On Uncertainty, Tempering, and Data Augmentation in Bayesian Classification
>
> [20]: Titsias, (2016). One-vs-Each Approximation to Softmax for Scalable Estimation of Probabilities
>
> [21]: Wightman et al., (2021). ResNet strikes back
>
> [22]: Li et al., (2025). Binary Cross-Entropy Loss Leads to Better Calibration than Focal Loss on Imbalanced Multi-Class Datasets
>
> [23]: Jürgens et al., (2024). The Epistemic Uncertainty of Evidential Deep Learning is not Faithfully Represented
>
> [24]: Pandey & Yu, (2023). Learn to Accumulate Evidence from All Training Samples: Theory and Practice
>
> [25]: Mucsányi et al., (2024). Benchmarking Uncertainty Disentanglement
>
> [26]: Maddison et al., (2014). A* Sampling
>
> [27]: Jang et al., (2017). Categorical Reparameterization with Gumbel-Softmax
>
> [28]: Maddison et al., (2017). The Concrete Distribution
>
> [29]: Huijben et al., (2023). A Review of the Gumbel-max Trick

---

> > ### Comment · Reviewer_LBev · 2025-08-04
> >
> > Thank you for your rebuttal and the additional results, I will keep my score.

---

### Official Review · Reviewer_bh5k · 2025-07-04

**Clarity:** 3
**Significance:** 3
**Originality:** 2
**Rating:** 4
**Confidence:** 3

**Summary:**

The paper "re-studies" supervised classification problems from the perspective of softmax functions and other transformations from the *logit* space. Based on concerns around the way of modelling epistemic/aleatoric uncertainty, and the memory cost required for using Monte Carlo approximations to obtain predictive probabilities in the standard way --- the authors introduce closed-form predictives and other distributional approximations. Sadly, the work ignores a central line of research in ML, which is the Gumbel-Max trick, the Gumbel-Softmax trick, and all other contributions that in the period 2014-2020 (approx) did great advances in problems related to sampling from discrete distributions or dealing with their log-partition functions.

**Questions:**

Could the authors explain in some detail why the Gumbel-softmax trick or similar methods are not of interest here? Or why should they not be considered? Any connection or linking with them would be really helpful for this reviewer, who is actually a bit lost on "why this work does not align with the references mentioned before"

In this direction, I would also like to point out the first 3 paragraphs in Section 2 of Maddison et al. (NIPS 2014).

This TPAMI review can also help to find some central references:

- Huijben, I. A., Kool, W., Paulus, M. B., & Van Sloun, R. J. (2022). A review of the gumbel-max trick and its extensions for discrete stochasticity in machine learning. _IEEE transactions on pattern analysis and machine intelligence_, _45_(2), 1353-1371.

**Ethical Concerns:**

["NO or VERY MINOR ethics concerns only"]

**Final Justification:**

The rating scored was updated from 3: borderline reject to > 4: borderline accept. I added some comments in my first response message to the authors' rebuttal.

**Quality:**

2

**Strengths And Weaknesses:**

First of all, I have to say that I am surprised that the work seems to ignore and does not review classic methods to deal with classification. I was for instance wondering about the lack of reference to the following works:

- Maddison, C. J., Tarlow, D., & Minka, T. (2014). A* sampling. NIPS.
- Jang, E., Gu, S., & Poole, B. (2016). Categorical reparameterization with Gumbel-softmax. _arXiv preprint arXiv:1611.01144_
- C. J. Maddison, A. Mnih, and Y. W. Teh, “The Concrete Distribution: A Continuous Relaxation of Discrete Random Variables,” 5th Int. Conf. Learn. Represent. ICLR 2017

From my point of view, the authors identify certain problems in the standard use of the softmax function and how it may affect certain predictive quantities, which are obviously related to the problem of sampling from a distribution (this is somehow described in the first paragraphs of pp. 2). In addition to the reasoning about aleatoric/epistemic uncertainty, which I think does not completely fit well here (for instance, the sentence starting in L30), they observe the general property of the logits before being used in the one-hot encoding.

The situation, therefore, is the following: Eq. 7 and Eq. 8 are presented as the main discovery or method to use, where it is a Gaussian expectation of the function $\exp(y)$. I see the advantage of the analytical result, but I would be very surprised if this has not been considered before or studied.
This result is then pushed forward and used to make the main approximation in Eq. 10, whose contribution I don't understand very well, or at least I question myself about the convenience of the expectation on top of the one-hot-encoding fraction. Are we really allowed to do this approximation? Seems extreme in my opinion, some discussion could be perhaps helpful.

In that direction, the absence of reference to log-partition functions, the log-sum-exp operator, and other (very) well-known methods makes me feel lost, or at least very disconnected from concepts and principles that (as a reader and practitioner) I know and have used quite often. So I cannot judge the importance of the method in a good way without such links.

After this central result on pp. 3 is included, the same sort of technique is reused two to three times, and the Gaussian integral of the exp() function and other simple functions is similarly considered in the Section 3. I reviewed experiments as well, but I cannot really value them well without first understanding the connections and solving the previous concerns and issues first.

---

> ### Author Rebuttal · Authors · 2025-07-31
>
> We thank the reviewer for their careful review of our manuscript.
>
> **Contextualisation.** We agree that better contextualization is needed. To this end, we have added a new Related Work section and an explicit list of contributions. To avoid repetition, these appear in full in our response to Reviewer `LBev`.
>
> **The Gumbel-max trick.** Thank you for highlighting this important related work. We have added all four of your suggested references of the Gumbel-max trick and related continuous relaxations to our Related Work section.
> While both our work and the Gumbel-max trick involve alternatives to standard softmax, they address different challenges. The Gumbel-max trick enables gradient-based optimization through discrete random variables by providing continuous relaxations of categorical sampling. Our approach instead provides fully _sample-free_ inference by deriving closed-form expressions for both predictives and second-order distributions over the probability simplex through moment matching. Among other things, our analytic approximate push-forward is radically faster than sampling (see also the response to Reviewer `a52w` below). The key difference is the _source_ of stochasticity. In Gumbel-max, the randomness originates from sampling discrete categories to enable gradient-based optimisation. In our setting, the randomness comes from Gaussian distributions over logits for _uncertainty quantification_, which we refer to as 'approximate Gaussian inference'. The prototypical example of an approximate Gaussian inference method is the Laplace approximation (e.g., [1]). This is an approximate Bayesian method that approximates the posterior over the neural network parameters with a Gaussian distribution. This Gaussian is then pushed forward to logit space, and the present work addresses how to further push this Gaussian forward onto the probability simplex *without* sampling.
>
> **Used approximations.** Thank you for examining our approximations carefully. To the best of our knowledge, Eqs. (7) and (8) have never been used for analytically approximating predictives. Importantly, Eqs. (7) and (8) are presented as a bridge between traditional softmax-based approximate Gaussian inference and our proposed predictive approximation for the sigmoid and normCDF activations, which we present as the main method to use. While the probit approximation has been used in _binary_ classification, we extend it to the multi-class case. We make this contribution clearer in our new list of contributions, which we display in full in our response to Reviewer `LBev`.
>
> In Eq. (12), the expectation of the fraction is approximated by the fraction of the expectation. Such a type of approximation is well-known in the Bayesian literature and sometimes referred to as a mean-field approximation [2], though in this work, we refer to the mean-field approximation as a specific one that has been proposed to approximate the Gaussian softmax integral following Lu et al. [3]. We discuss the quality of our approximation later in the paper, both theoretically (Theorem 2.1) and empirically (in the synthetic experiment, Figure 1). Crucially, this approximation is highly accurate with our proposed normCDF and sigmoid activations, as shown in our experiments. As the variance across the different classes decreases, the approximation becomes more accurate (in the limiting deterministic case, the approximation is exact). The aim of Theorem 2.1 is to quantify this limiting behaviour, which we believe is the first error analysis for such a mean-field approximation in probabilistic deep learning.
>
> We hope that our new Related Work section, the list of contributions, and clarifications above address the reviewer's concerns. We are happy to clarify any remaining points.
>
> [1]: Daxberger et al., (2021). Laplace Redux -- Effortless Bayesian Deep Learning
>
> [2]: Friedman & Koller, (2009). Probabilistic Graphical Models -- Principles and Techniques
>
> [3]: Lu et al., (2021). Mean-field approximation to Gaussian-Softmax integral with application to uncertainty estimation

---

> > ### Comment · Reviewer_bh5k · 2025-08-06
> >
> > As far as I can see, the authors did a good job responding to my comments and also in their rebuttal for the rest of the reviews. I am particularly happy with the information and answers provided, as they are informative and clarify quite a lot. I don't have additional questions, so that's one of the reasons why I did not engage before. In some sense, it seems that this rebuttal here on the _openreview_ platform contains better context and clarity than some parts of the submitted manuscript... In that way, it would be great to see all of this incorporated in the revised version --- but as doing so can be considered like _major updates_, I am not sure if it makes sense to go for an acceptance score here.
> >
> > In that direction, I am also aligned with **Rev uK2i** as I can't consider myself an expert on the exact topic of mappings for classification models. For that reason, I trust somehow the decisions taken by **Revs LBev** and **a52w**, whose technical concerns were better "oriented" than the ones I raised.
> >
> > If there is a quorum for acceptance, I provide my support. However, I remark here that **my general preference** is to vote for a rejection now and hopefully have a resubmission in the near future, incorporating all the feedback from this reviewing phase (which for sure would improve the impact of the work!)

---

### Decision · Program_Chairs · 2025-09-17

**Decision:**

Accept (poster)

**Comment:**

The reviewers of this submission all recommend borderline acceptance. Some of the reasons for these positive evaluations include writing quality (Reviewer LBev), studying a problem which is common-place but rarely explicitly considered specifically analytic integration of softmax-type expressions as opposed to sampling (Reviewer a52w, Reviewer uK2i), and good performance improvements on a set of reasonable baselines (Reviewer uK2i). Weaknesses include a lack of discussion of relevant prior work. I would be very surprised if the authors did not know of this work because some of the mentioned papers on Gumbel-Max extensions are very widely known, and think a better explanation is that the authors are aware of the work but simply didn't quite draw the connection due to their different goals. I think this weakness is minor enough to be addressable at a camera-ready stage, and would encourage the authors to indeed add the requested references mentioned by Reviewer bh5k. I did not detect any other major weaknesses, but on the other hand also think the paper's merits - especially the potentially very broad potential applicability of the ideas given ubiquity of softmax scores both in the approximate inference world and in deep learning more broadly - are sufficient for NeurIPS.

On basis of reviewer consensus, I recommend the work is accepted.